# Cortical ChAT+ neurons co-transmit acetylcholine and GABA in a target- and brain-region-specific manner

Adam J Granger[1], Wengang Wang[1], Keiramarie Robertson[1], Mahmoud El-Rifai[2], Andrea F Zanello[1], Karina Bistrong[1], Arpiar Saunders[3], Brian W Chow[4], Vicente Nuñez[4], Miguel Turrero García[4], Corey C Harwell[4], Chenghua Gu[4], Bernardo L Sabatini[1]*

[1]Howard Hughes Medical Institute, Department of Neurobiology, Harvard Medical School, Boston, United States; [2]Neurobiology Imaging Facility, Department of Neurobiology, Harvard Medical School, Boston, United States; [3]Department of Genetics, Harvard Medical School, Boston, United States; [4]Department of Neurobiology, Harvard Medical School, Boston, United States

**Abstract** The mouse cerebral cortex contains neurons that express choline acetyltransferase (ChAT) and are a potential local source of acetylcholine. However, the neurotransmitters released by cortical ChAT+ neurons and their synaptic connectivity are unknown. We show that the nearly all cortical ChAT+ neurons in mice are specialized VIP+ interneurons that release GABA strongly onto other inhibitory interneurons and acetylcholine sparsely onto layer 1 interneurons and other VIP+/ ChAT+ interneurons. This differential transmission of ACh and GABA based on the postsynaptic target neuron is reflected in VIP+/ChAT+ interneuron pre-synaptic terminals, as quantitative molecular analysis shows that only a subset of these are specialized to release acetylcholine. In addition, we identify a separate, sparse population of non-VIP ChAT+ neurons in the medial prefrontal cortex with a distinct developmental origin that robustly release acetylcholine in layer 1. These results demonstrate both cortex-region heterogeneity in cortical ChAT+ interneurons and target-specific co-release of acetylcholine and GABA.

*For correspondence:
bsabatini@hms.harvard.edu

**Competing interests:** The authors declare that no competing interests exist.

## Introduction

Acetylcholine (ACh) is a neurotransmitter and neuromodulator that is released throughout the mammalian cortex at times of alertness and arousal (*Teles-Grilo Ruivo et al., 2017*) in order to promote learning and memory (*Hasselmo, 2006*), modulate sensory perception (*Pinto et al., 2013*), gate plasticity (*Morishita et al., 2010*; *Rasmusson, 2000*), and enhance the detection of salient sensory cues and reinforcement (*Parikh et al., 2007*; *Sarter et al., 2009*; *Sarter et al., 2014*; *Sturgill et al., 2020*). Most cortical ACh originates from subcortical nuclei in the basal forebrain *Mesulam, 1995* whose long-range axons innervate broad regions of cortex and release ACh to modulate cortical function over fast and slow time scales (*Sarter et al., 2009*; *Picciotto et al., 2012*). However, cholinergic interneurons are present in the cortex of mice and rats and could provide a local source of ACh. Unfortunately, the physiology and function of these cells are poorly understood and their contribution to cortical signal has been controversial.

Putative cholinergic neurons in the cortex were first identified by immunolabeling for choline acetyltransferase (ChAT), the biosynthetic enzyme that produces ACh (*Eckenstein and Thoenen, 1983*; *Eckenstein and Baughman, 1984*), and their presence has since been corroborated by both immunohistochemical and transcriptional analyses (*Bhagwandin et al., 2006*; *Cauli et al., 2014*; *Consonni et al., 2009*; *Gonchar et al., 2007*; *Kosaka et al., 1988*; *Peters and Harriman, 1988*;

*Porter et al., 1998*; *Schäfer et al., 1994*; *Weihe et al., 1996*; *Chédotal et al., 1994*). Both initial immunochemical labeling and recent single-cell transcriptomic classification *Saunders et al., 2018*; *Tasic et al., 2016*; *Zeisel et al., 2015* demonstrate that cortical ChAT+ neurons also express vasoactive intestinal peptide (VIP), indicating that they are a subclass of VIP+ interneurons.

To date, characterization of the synaptic connectivity of cortical ChAT+ neurons has been limited, describing primarily cholinergic effects on downstream neurons, with little or no GABAergic effects as might be expected from a subclass VIP+ interneurons. Von Engelhardt and colleagues reported that cortical ChAT+ neurons release ACh that opens nicotinic ACh receptors (nAChRs) on excitatory pre-synaptic terminals to increase synaptic release of glutamate (*von Engelhardt et al., 2007*). A recent study by Obermayer et al found that cortical ChAT+ neurons directly excite several interneuron subtypes as well as deep layer pyramidal neurons via nAChRs (*Obermayer et al., 2019*). These studies argue strongly for a primarily cholinergic role for these neurons. However, they did not comprehensively survey post-synaptic connectivity across cortex.

Whether cortical ChAT+ neurons also release gamma-aminobutyric acid (GABA) is even less clear. Several studies have reported GABA synthetic enzyme expression in only a subset of cortical ChAT+ neurons (*Kosaka et al., 1988*; *von Engelhardt et al., 2007*), whereas others have reported widespread co-labeling with GABA (*Bayraktar et al., 1997*). Although the synaptic outputs of cortical ChAT+ neurons have either been described as entirely cholinergic (*von Engelhardt et al., 2007*) or partially GABAergic (*Obermayer et al., 2019*), activation of cortical ChAT+ neurons in vivo suppresses responses to sensory input (*Dudai et al., 2020*). This could occur either through directly via GABAergic inhibition or indirectly by cholinergic excitation of intermediate inhibitory interneurons. Given their expression of VIP, a marker gene for a cardinal class of GABAergic interneurons, one would expect cortical ChAT+ neurons to be GABAergic, but this has not been definitively shown.

We previously reported that co-transmission of GABA is a common feature of cholinergic neurons in the mouse forebrain (*Saunders et al., 2015a*; *Granger et al., 2016*; *Saunders et al., 2015b*). Because GABA and ACh have opposite effects on membrane voltage through ionotropic receptors, the functional consequences of their co-transmission on cortical circuits is unknown. One possibility is that they each transmit onto the same post-synaptic targets and have competing effects, similar to the co-release of GABA and glutamate in the habenula from entopeduncular neurons (*Shabel et al., 2014*; *Wallace et al., 2017*) or co-release of ACh and GABA from starburst amacrine cells onto direction-selective retinal ganglion cells (*Lee et al., 2010a*; *Sethuramanujam et al., 2016*). Another possibility is that they target different post-synaptic cells, which could allow them to have complementary network effects. To differentiate between these alternatives requires determining the molecular competency of cortical ChAT+ neurons to release GABA and ACh from their pre-synaptic terminals, and systematic examination of their synaptic connectivity.

To answer these many unknowns, we molecularly and functionally characterized cortical ChAT+ neurons and describe two classes of cortical ChAT+ neurons. The first is a subset of VIP+ interneurons, and expresses the necessary cellular machinery to synthesize and release both ACh and GABA. A systematic survey of synaptic connectivity shows that, for these cells, most synaptic output is GABAergic. Specifically, GABA release is robust onto somatostatin (Sst)-expressing interneurons, similar to the larger population of VIP+ interneurons. However, these cells are capable of releasing ACh, with sparse and highly specific targeting of ACh mostly onto layer 1 interneurons and other cortical VIP+/ChAT+ neurons. Target-specificity is partially specified at the pre-synaptic level, as we identified two distinct populations of pre-synaptic terminals: a subset that are competent to release both GABA and ACh and others that can only release GABA. The second class of cortical ChAT+ interneurons is molecularly and functionally distinct from VIP+ cholinergic interneurons and was discovered in an effort to reconcile our results with those of another study that described predominantly ACh, and not GABA release, from cortical ChAT+ interneurons (*Obermayer et al., 2019*). This sparse population of non-VIP ChAT+ neurons is found in the mPFC, has a distinct developmental origin from VIP+ interneurons, and contributes primarily cholinergic signaling. Thus, ChAT+ interneurons are heterogeneous across cortical regions, comprise an intra-cortical source of highly specific synaptic ACh, and show target-specific co-transmission of two distinct neurotransmitters.

## Results

### Cortical VIP⁺/ChAT⁺ neurons express genes for release of both ACh and GABA

To visualize potential cholinergic neurons in the cortex, we genetically labeled all *Chat*-expressing cells with tdTomato (*Chat^{ires-Cre}* x *Rosa26^{lsl-tdTomato}*), and observed putative cholinergic neurons throughout the cortex (*Figure 1A*). We confirmed that *Cre* expression faithfully reports *Chat* expression in cerebral cortex using fluorescent in situ hybdrization (FISH), with 97% of *Chat*⁺ neurons expressing *Cre* and 100% of *Cre*⁺ neurons expressing *Chat* (*Figure 1B*). In contrast, a population of neurons in the subiculum are also strongly labeled in *Chat^{ires-Cre}* x *Rosa26^{lsl-tdTomato}* mice (*Figure 1A*), but do not express *Chat* in the adult (data not shown). In addition to *Chat,* neurons also require the expression of the membrane choline transporter, encoded by *Slc5a7*, and the vesicular ACh transporter (VAChT), encoded by *Slc18a3,* to synthesize and release ACh. Both of these genes are also expressed in the majority of cortical ChAT⁺ neurons (*Figure 1C,D*), indicating that cortical ChAT⁺ neurons have all the molecular machinery necessary to release ACh. These neurons display a vertically-oriented morphology, with their main dendrites aligned perpendicular to the cortical surface, and are either bipolar, with two main vertical dendrites (*Figure 1E*, 66% of all cortical ChAT⁺ neurons) or multipolar, with three or more main dendrites (*Figure 1E*, 34% of all cortical ChAT+ neurons). They cluster in superficial layers, especially near the border between layers 1 and 2 (*Figure 1F*).

Previous studies have reported conflicting results on the extent to which these neurons are GABAergic, and they are often shown to co-label with vasoactive intestinal peptide (VIP) (*Eckenstein and Baughman, 1984*). We confirmed using both immunohistochemistry and FISH that cortical ChAT⁺ neurons comprise an ~33% subset of VIP⁺ interneurons (*Figure 2A,B*), and do not co-label with either parvalbumin (PV) or somatostatin (Sst, *Figure 2—figure supplement 1*). To test whether cortical ChAT⁺ neurons are able to release GABA, we performed FISH for the GABA handling and synthesis genes *Slc32a1*, encoding the vesicular GABA transporter (VGAT), and *Gad1,2*, which encode the GABA synthetic enzymes. Nearly all cortical ChAT⁺ neurons express both *Slc32a1* and *Gad1*,2 (*Figure 2C*). These results are corroborated by single-cell RNA sequencing data from the Allen Institute (*Tasic et al., 2016*), indicating that a subset of *Vip*-expressing cortical interneurons also express cholinergic genes *Chat, Slc5a7, Slc18a3*, and GABAergic genes *Slc32a1, Gad1,* and *Gad2*, but not glutamatergic genes (*Figure 2—figure supplement 2*). In sum, these data show that cortical VIP⁺/ChAT⁺ interneurons have the potential for synaptic release of both ACh and GABA.

### Cortical VIP⁺/ChAT⁺ neurons robustly release GABA onto inhibitory interneurons and sparsely release ACh

To confirm which neurotransmitters VIP⁺/ChAT⁺ neurons release and understand the circuit function these different neurotransmitters provide, we electrophysiologically screened for the post-synaptic output of cortical ChAT⁺ neurons. In order to identify synaptic outputs, as opposed to possible effects of volume transmission, we focused on synaptic effects mediated by activation of post-synaptic ionotropic receptors. We virally delivered Cre-dependent ChR2-mCherry (AAV(8)-DIO-ChR2-mCherry) into the motor cortex of *Chat^{ires-Cre}* mice, and allowed three weeks for viral gene expression, prepared acute brain slices and recorded whole-cell voltage clamp responses from ChR2-lacking neurons while stimulating nearby ChR2-expressing neurons with blue light (*Figure 3A*). We screened for post-synaptic responses mostly in primary motor cortex (M1), with some recordings in visual cortex (V1). Because we saw no differences in connectivity between these two regions, we have pooled that data here. Synaptic responses mediated by nicotinic ACh receptors (nAChRs) were identified by voltage clamping the post-synaptic neurons at −70 mV in the presence of NBQX to preclude any contamination by feed-forward glutamatergic currents, and by sensitivity to nAChR-selective antagonists (*Figure 3B*). We observed nAChR-mediated responses with both slow and fast components (*Figure 3B*), as well as several with only fast components (not shown), indicating variability in the nAChR receptor composition in post-synaptic neurons (*Bennett et al., 2012*). GABA_AR-mediated synaptic currents were identified by voltage clamping the cell at 0 mV, and by blocking with the GABA_AR-selective antagonist gabazine. We also confirmed that GABA responses were monosynaptic by sequential block with TTX and rescue by 4AP (*Figure 3C*; *Petreanu et al., 2009*),

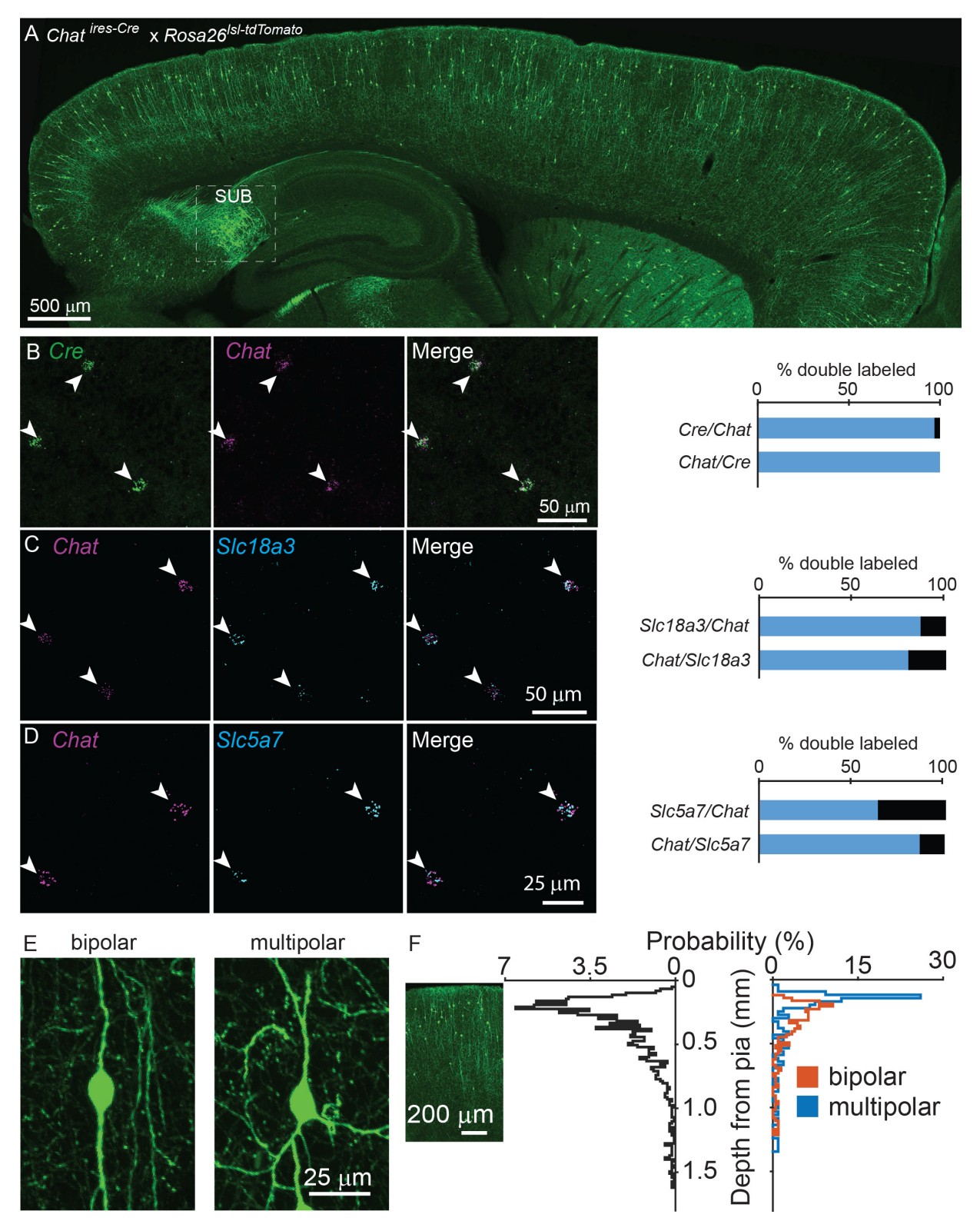

**Figure 1.** Cortical ChAT+ neurons are present throughout cortex and express genes necessary for synthesis and release of ACh. (**A**) Sagittal view of a mouse neocortex with ChAT+ neurons expressing tdTomato (*Chat^ires-Cre* x *Rosa26^lsl-tdTomato*) demonstrating the distribution of putative cholinergic neurons throughout the cortex. Strongly tdTomato-labeled neurons in the subiculum that do not express *Chat* are labeled (SUB). (**B**) Flourescent in situ hybridization of *Cre* faithfully reports *Chat* expression in *Chat^ires-Cre* mice in the cortex. Arrow heads indicate dual *Chat+/Cre+* neurons. Quantification

*Figure 1 continued on next page*

*Figure 1 continued*

shown at right (n = 32 *Chat*⁺/*Cre*⁺ of 33 *Chat*⁺ and 32 *Cre*⁺ neurons from 2 *Chat*^{ires-Cre} mice). (**C,D**) Fluorescent in situ hybrization of *Chat* in cortex co-labels with *Slc18a3*, the gene encoding VAChT (n = 147 *Chat*⁺/*Slc18a3*⁺ of 170 *Chat*⁺ and 184 *Slc18a3*⁺ neurons from 3 wild-type mice) and *Slc5a7*, the gene encoding the membrane choline transporter (n = 72 *Chat*⁺,*Slc5a7*⁺ of 113 *Chat*⁺ and 83 *Slc5a7*⁺ neurons from 3 wild-type mice). Arrowheads indicate cortical ChAT⁺ neurons and quantification shown at right. (**E**) Cortical ChAT⁺ neurons are vertically oriented and are bipolar (left) or multipolar (right). (**F**) Distribution of cortical depth from the pia of all cortical ChAT⁺ neurons (left graph, black trace, n = 1059 neurons from 3 *Chat*^{ires-Cre} x *Rosa26*^{lsl-tdTomato} mice), median cell body is 274 µm from pia ±15 µm, 95% C.I.) and according to morphology (right graph; orange = bipolar, n = 207, 66% of total, median 293 µm from pia ±23 µm, 95% C.I.; blue = multipolar, n = 107 neurons, 34% of total, median 173 µm from pia ±24 µm, 95% C.I.). Inset image is aligned to the relative depth shown in the graphs.

confirming they were not the result of indirect excitation of intermediate inhibitory neurons. Of the neurons that displayed a detectable synaptic response following optogenetic stimulation of the cortical ChAT⁺ neurons, most showed a GABA_AR-mediated current, confirming that cortical ChAT⁺ neurons are indeed GABAergic. A smaller subset of neurons in superficial layers showed nAChR-mediated synaptic responses (*Figure 3D*). All but two (of 49) responsive neurons displayed either GABA_AR- or nAChR-mediated currents, not both, indicating that the synaptic release of GABA or ACh by cortical ChAT⁺ neurons is independent and differentially targeted based on the output neurons.

To identify onto which neuron populations cortical ChAT⁺ neurons synapse and therefore inform the potential circuit function of both ACh and GABA release, we systematically surveyed connectivity to specific neuronal subtypes. We repeated the ChR2-assisted connectivity survey described above,

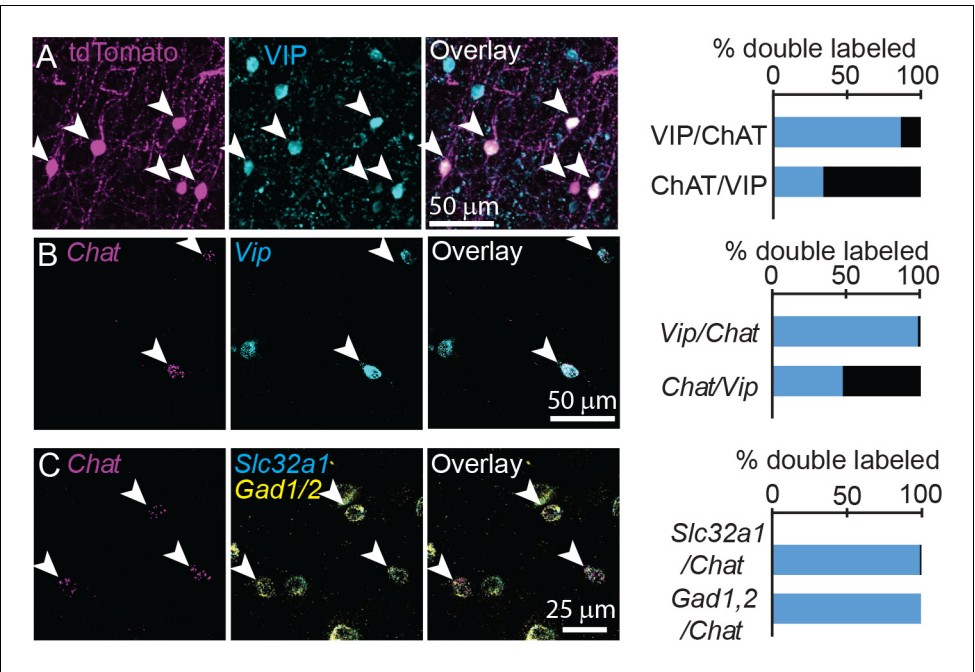

**Figure 2.** Cortical ChAT⁺ neurons are a subset of VIP⁺ interneurons and express genes necessary for synthesis and release of GABA. (**A**) Cortical ChAT⁺ neurons expressing tdTomato (*Chat*^{ires-Cre} x *Rosa26*^{lsl-tdTomato}) co-label with immunostained VIP (n = 127 ChAT⁺/VIP⁺ neurons of 147 total ChAT⁺ and 375 VIP⁺ neurons from 3 *Chat*^{ires-Cre} x *Rosa26*^{lsl-tdTomato} mice). (**B**) Fluorescent in situ hybridization of *Chat* in cortex co-labels with *Vip* (n = 278 *Chat*⁺/ *Vip*⁺ of 283 *Chat*⁺ and 579 *Vip*⁺ neurons from 3 wild-type mice). (**C**) Fluorescent in situ hybridization labeling of *Chat* in cortex co-labels with the GABAergic genes *Slc32a1*, which encodes for VGAT, and *Gad1* and *Gad2*, which encodes for the GABA synthetic enzymes GAD67 and GAD65, respectively (n = 101 *Chat*⁺,*Slc32a1*⁺ and 102 *Chat*⁺,*Gad1/2*⁺ of 102 *Chat*⁺ neurons from 5 wild-type mice). Arrowheads indicate double labeled neurons. The online version of this article includes the following figure supplement(s) for figure 2:

**Figure supplement 1.** Cortical ChAT⁺ neurons do not express Parvalbumin or Somatostatin.

**Figure supplement 2.** A subset of VIP⁺ interneurons express cholinergic genes by single-cell RNA sequencing.

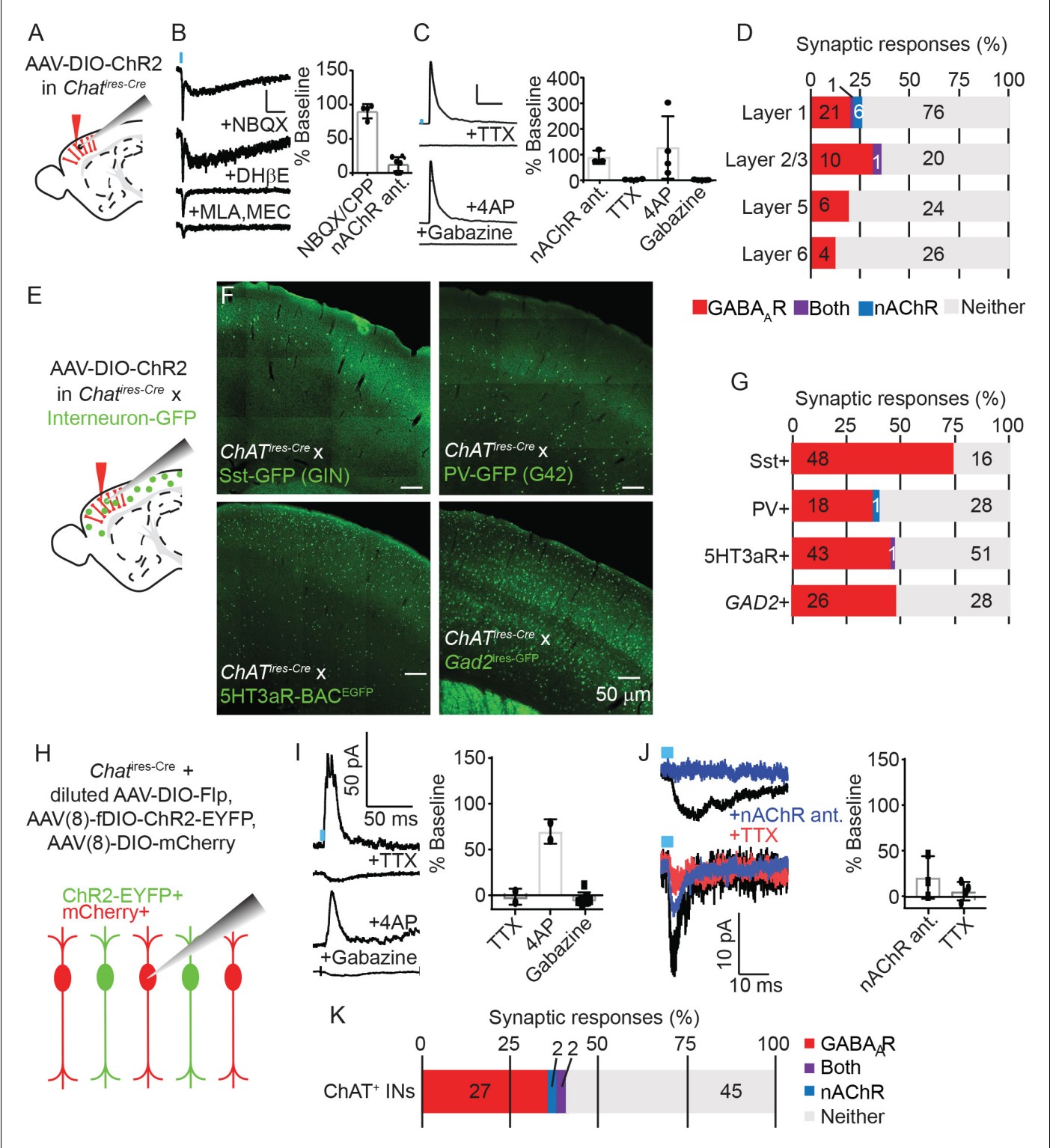

**Figure 3.** Cortical VIP+/ChAT+ interneurons primarily release GABA onto inhibitory interneurons and sparsely release ACh onto layer 1 and other ChAT+ neurons. (**A**) Experimental design: Cre-dependent ChR2-mCherry is virally delivered (AAV(8)-DIO-ChR2-mCherry) to the cortex, expressed for 3 weeks, and whole-cell voltage clamp recordings obtained from unlabeled neurons. Recordings from motor cortex and visual cortex are pooled for panels **A-D**. (**B**) Examples trace of a biphasic nAChR-mediated synaptic currents isolated by voltage clamping the post-synaptic neuron at −70 mV and stimulating cortical ChAT+ neurons with 3 ms of 473 nm light (~7–9 mW/cm²). The synaptic current is insensitive to AMPA receptor antagonist NBQX, but the slow component is blocked by DHβE, selective for α4 receptor subunits, and the fast component is blocked by the α7-selective antagonist MLA
*Figure 3 continued on next page*

*Figure 3 continued*

and pan-nAChR antagonist MEC. Right panel shows summary quantification of sensitivity to glutamatergic antagonists NBQX/CPP and nAChR antagonists DHβE, MLA, and MEC. Error bars show mean +/- s.e.m. (C) Example GABA$_A$ receptor-mediated currents isolated by voltage clamping the post-synaptic neuron at 0 mV. Synaptic currents are blocked by voltage-gated sodium channel antagonists TTX, rescued by subsequent application of potassium channel antagonists 4AP, and further blocked by GABA$_A$R-selective antagonist gabazine. Right panel shows summary quantification of the effects of nAChR antgonists, TTX, 4AP, and gabazine on inhibitory currents. (D) Summary of the proportion of neurons showing synaptic responses following optogenetic stimulation of cortical ChAT$^+$ neurons across cortical layers. The numbers in each bar indicate the number of cells in each category. (n = 195 total neurons from 24 *Chat$^{ires-Cre}$* mice; p=0.8627 for layer 1 compared to non-layer 1 GABA$_A$R-mediated responses and p=0.0695 or layer compared to non-layer1 nAChR-mediated responses, Fisher's exact test). (E) Experimental design: AAV(8)-DIO-ChR2-mCherry was injected into the motor cortex of *Chat$^{ires-Cre}$* mice crossed to different mouse lines that express GFP in specific interneuron subpopulations. (F) Example images showing GFP expression in 4 different mouse lines expressing GFP in different interneuron subtypes. (G) Summary quantification of the proportion of cells of each interneuron subtype that had synaptic responses to optogenetic stimulation of cortical ChAT$^+$ neurons. The numbers in each bar indicate the number of cells in each category. (n = 64 GFP$^+$ cells from 6 *Chat$^{ires-Cre}$* x Sst-GFP (GIN) mice; n = 47 GFP$^+$ cells from 4 *Chat$^{ires-Cre}$* x PV-GFP (G42) mice; n = 95 GFP$^+$ cells from 6 *Chat$^{ires-Cre}$* x 5HT3aR-BAC$^{EGFP}$ mice; n = 54 GFP$^+$ cells from 4 *Gad2$^{ires-GFP}$* mice; p=0.0307 for differences in GABA$_A$R-responses between interneuron types, p=0.5775 for differences in nAChR-responses, Pearson's chi-squared test). (H) Experimental design: To achieve mosaic expression of ChR2 in a subset of cortical ChAT$^+$ neurons, we injected a diluted AAV(8)-DIO-FlpO virus so that a subset would express Flp. We then injected with high-titer AAV(8)-fDIO-ChR2-EYFP and AAV(8)-DIO-mCherry. We targeted mCherry$^+$, EYFP$^-$ cells for whole-cell voltage clamp recording that neighbored EYFP$^+$ neurons. (I) Example traces showing putative GABA$_A$R-mediated synaptic response at baseline and following application of TTX, 4AP, and gabazine. Right panel shows summary quantification of block by TTX, rescue by 4AP, and block by gabazine. Error bars show mean ± s.e.m. (J) Example traces of two different neurons showing nAChR-mediated responses and their block by nAChR antagonists and TTX. These two cells have different response kinetics, potentially indicative of extra-synaptic (top) and synaptic (bottom) nAChRs. Right panel shows summary quantification of putative nAChR-mediated synaptic response sensitivity to nAChR antagonists (MEC, MLA, and DHβE) and TTX. (K) Summary quantification of the proportion of cortical ChAT$^+$ neurons that showed synaptic responses following stimulation of neighboring ChR2-expressing cells. The numbers in each bar indicate the number of cells in each category (n = 76 neurons from 8 *Chat$^{ires-Cre}$* mice).

The online version of this article includes the following figure supplement(s) for figure 3:

**Figure supplement 1.** GABA$_A$R-mediated synaptic currents from VIP$^+$/ChAT$^+$ interneurons are mono-synaptic.
**Figure supplement 2.** Cortical VIP$^+$/ChAT$^+$ neurons have a low rate of connectivity to putative pyramidal neurons.
**Figure supplement 3.** Cortical VIP$^+$/ChAT$^+$ neurons inhibit deep layer Sst$^+$ interneurons.
**Figure supplement 4.** Cortical VIP$^+$/ChAT$^+$ interneuron have no effect on spontaneous excitatory post-synaptic currents in pyramidal neurons.
**Figure supplement 5.** ACh release from VIP$^+$ interneurons is not necessary for neurovascular coupling.

but in *Chat$^{ires-Cre}$* mice crossed with transgenic lines that express GFP in the major interneuron populations, including Sst$^+$ (*Oliva et al., 2000*), PV$^+$ (*Chattopadhyaya et al., 2004*), and 5HT3aR$^+$ interneurons (*Lee et al., 2010b*; *Figure 3E,F*). We found high rates of GABAergic connectivity, especially onto Sst$^+$ interneurons, while nAChR-mediated responses were rare (*Figure 3G*). While Sst$^+$, PV$^+$, and 5Ht3aR$^+$ interneurons represent nearly 100% of all cortical interneurons (*Rudy et al., 2011*), the Sst- and PV-labeling transgenic lines incompletely label their respective interneuron populations. We therefore also recorded responses from GFP-labeled, GAD65-expressing interneurons from *Gad2$^{ires-GFP}$* mice, and observed only GABA$_A$R-mediated responses (*Figure 3G*). In each of these specific neuronal subtypes, we confirmed that GABA release from cortical ChAT$^+$ cells was monosynaptic and confirmed that between these 4 interneuron classes, we spanned the entire cortical column (*Figure 3—figure supplement 1*). We also targeted pyramidal neurons based on their morphology and laminar position and found a low overall rate of connectivity, which was entirely GABA$_A$R-mediated (*Figure 3—figure supplement 2*). This pattern of connectivity is consistent with reports for VIP$^+$ interneurons as a whole (*Pfeffer et al., 2013*; *Karnani et al., 2016a*), and indicates that the main circuit function of cortical VIP$^+$/ChAT$^+$ interneurons is disinhibition.

While this broad connectivity survey makes clear that VIP$^+$/ChAT$^+$ neurons release GABA most robustly onto Sst$^+$ interneurons, it is less clear exactly which neurons receive nAChR-mediated input. A subset of layer 1 interneurons, whose specific molecular identity is otherwise unknown, showed the most robust ACh-mediated responses (*Figure 3D*). We therefore used a candidate-based approach to test specific potential post-synaptic populations that might be most likely to receive VIP$^+$/ChAT$^+$ input. A previous study has reported that non-Martinotti Sst$^+$ interneurons in layer 6 can be activated by muscarinic receptors in response to sensory stimulation (*Muñoz et al., 2017*), but the transgenic line we used to identify Sst$^+$ interneurons does not effectively label deep layer neurons (*Figure 3F*). We therefore recorded from deep-layer Sst$^+$ neurons by injecting *Chat$^{ires-Cre}$* x *Sst$^{ires-Flp}$* mice with viruses expressing Cre-dependent ChR2-mCherry (AAV(8)-DIO-ChR2-mCherry)

and Flp-dependent EYFP (AAV(8)-fDIO-EYFP), and obtained current clamp recordings to allow for detection of muscarinic currents following trains of optogenetic stimulation. Of those cells with clear synaptic responses, we only identified hyperpolarizing currents that were sensitive to gabazine, indicating they were GABA$_A$R-mediated (*Figure 3—figure supplement 3*). Contrary to the findings of von Engelhardt et al., we did not observe any significant effect of optogenetic stimulation on presynaptic glutamate release (*Figure 3—figure supplement 4*).

Given the ability of ACh to dilate blood vessels, and previous reports on the role of VIP$^+$ interneurons in mediating vasodilation (*Consonni et al., 2009*; *Chédotal et al., 1994*; *Kocharyan et al., 2008*), we also hypothesized that VIP$^+$/ChAT$^+$ interneurons might release ACh onto neighboring arteries, coupling an increase in cortical activity via disinhibition with an increase in blood flow to meet the increases in metabolic demand. Although we confirmed that optogenetic stimulation of VIP$^+$ interneurons is sufficient to induce vasodilation, using a genetic strategy that eliminates Ach release from VIP$^+$ interneurons, we found that ACh release from these cells is not necessary for optogenetic- or sensory-evoked vasodilation (*Figure 3—figure supplement 5*).

Finally, another study found that VIP$^+$ interneurons can increase their firing rate through cooperative excitation via nAChRs (*Karnani et al., 2016b*). We therefore devised a strategy to test for synaptic connectivity between VIP$^+$/ChAT$^+$ neurons by injecting *Chat$^{ires-Cre}$* mice first with a diluted Cre-dependent Flp virus, followed by high titer Flp-dependent ChR2-EYFP and Cre-dependent mCherry. We then recorded from mCherry-positive, EYFP-negative neurons while stimulating with blue light (*Figure 3H*). We found that VIP$^+$/ChAT$^+$ neurons largely release GABA onto each other (*Figure 3I, K*), but that a subset received nAChR-input which could be blocked by nAChR-selective antagonists (*Figure 3J,K*). These results demonstrate that VIP$^+$/ChAT$^+$ neuron output is primarily GABAergic, but is able to release ACh onto highly specific sub-networks of layer 1 interneurons and other VIP$^+$/ChAT$^+$ neurons.

## Cortical VIP$^+$/ChAT$^+$ pre-synaptic terminals are differentially enriched for GABA and ACh release machinery

Throughout the analysis of synaptic connectivity we found robust GABAergic currents in many neurons, and only relatively few cells with nicotinic receptor-mediated currents, even though many of the post-synaptic populations we examined express nAChRs. Several scenarios could explain this finding. One possibility is that most pre-synaptic terminals of VIP$^+$/ChAT$^+$ neurons are incapable of releasing ACh. Alternatively, most terminals might release both ACh and GABA, which would suggest that post-synaptic sites lack the nAChRs required to generate ionotropic currents following ACh release. To distinguish between these possibilities, we used array tomography to examine the pre-synaptic release machinery present in individual presynaptic terminals of cortical ChAT$^+$ interneurons in the motor cortex. We labeled the presynaptic terminals by injecting AAV-encoding Cre-dependent synaptophysin-YFP into the motor cortex of *Chat$^{ires-Cre}$* mice (*Figure 4A,B*) and analyzed the expression of seven synaptic proteins relative to YFP-labeled terminals. Specifically, we labeled for Synapsin 1 as a generic pre-synaptic marker, PSD-95 and VGLUT1 to label glutamatergic synapses, Gephyrin and VGAT to label GABAergic synapses, and ChAT and VAChT to label cholinergic synapses (*Figure 4C*). DAPI was also used to label nuclei.

We first analyzed this data by calculating the global cross-correlations of image intensity across all possible pairs of synaptic markers and DAPI to reveal the baseline level of colocalization (*Figure 4D*, also see *Micheva and Smith, 2007*). We also examined the colocalization of synaptic markers within motor cortex cortical ChAT$^+$ terminals, by calculating signal covariances specifically in the ~0.1% area of the images containing synaptophysin-YFP labeled pre-synaptic terminals (see Methods). Compared to the global cross-correlations (*Figure 4D*), this revealed high covariance of staining intensity for GABAergic and cholinergic markers, with little to no covariance with the glutamatergic markers (*Figure 4E*). Thus, the fluorescence of pre-synaptic markers of ACh and GABA release are correlated within the pre-synaptic terminals of cortical ChAT$^+$ neurons, indicating that these terminals have machinery to release both ACh and GABA, but not glutamate.

We also analyzed whether GABAergic and cholinergic proteins are enriched in terminals of cortical ChAT$^+$ interneurons in the motor cortex. To rigorously examine the enrichment of these synaptic proteins in cortical ChAT$^+$ terminals, we quantified z-scores that measured the enrichment of each pre-synaptic antibody marker relative to randomized controls (*Figure 4F*; further details on analysis in methods and *Figure 4—figure supplement 2*). Across all samples, Synapsin-1, Gephyrin, VGAT,

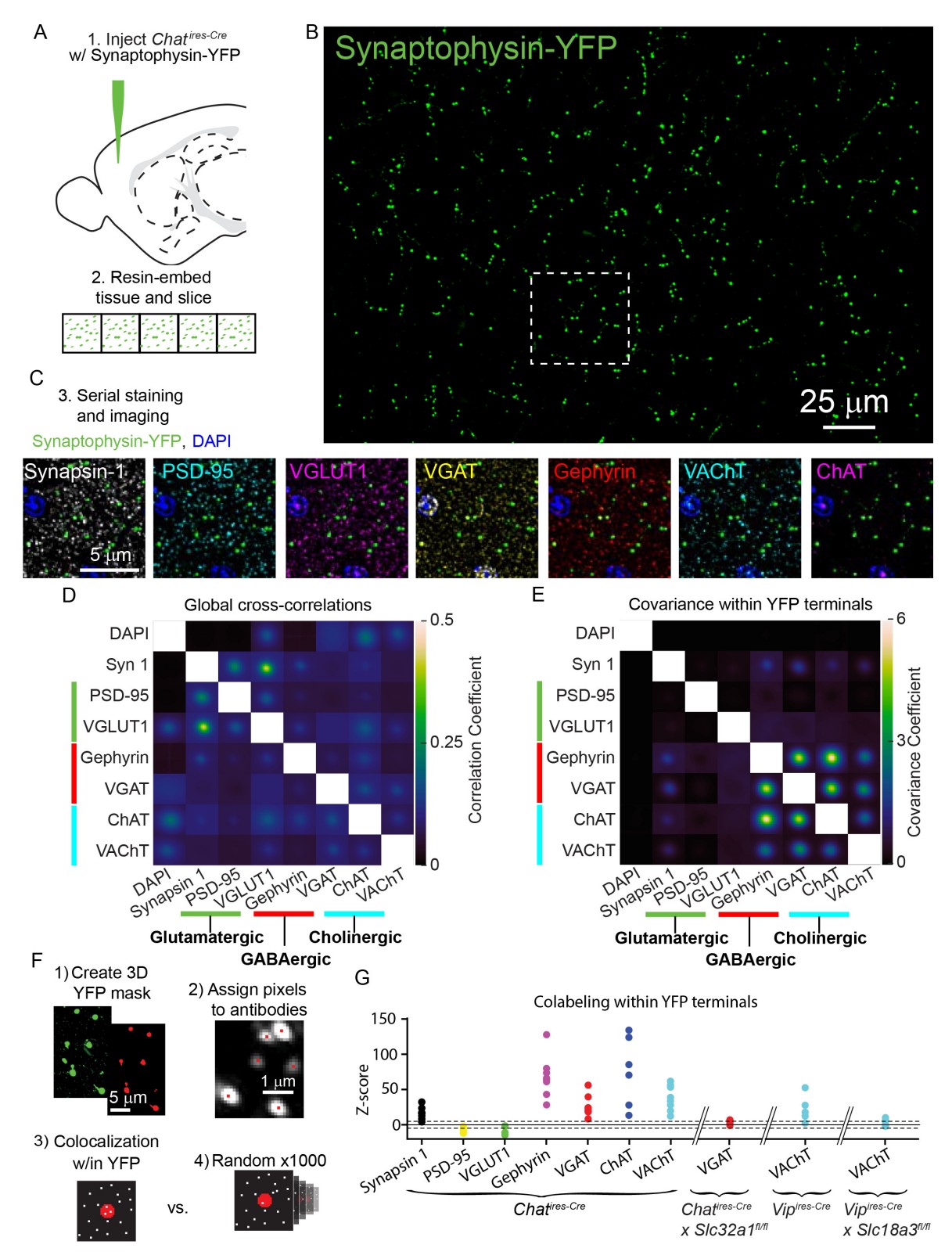

**Figure 4.** ACh and GABA synthesis and release machinery are both expressed in the presynaptic terminals of cortical ChAT[+] interneurons. (**A**) Array tomography workflow: *Chat^{ires-Cre}* mice are injected with Cre-dependent Synaptophysin YFP virus (AAV(8)-DIO-Synaptophysin-YFP) to label the pre-synaptic terminals of cortical ChAT[+] neurons (1).~1 mm[2] squares of tissue were then embedded in resin and cut into 70 nm slices in an array (2). (**B**) Example maximum-projection of Synaptophysin-YFP staining. (**C**) The ribbons of tissue are serially stained with antibodies against pre- and post-

*Figure 4 continued on next page*

Figure 4 continued

synaptic proteins (3). Example images show the inset from (B) to demonstrate the staining of pre-synaptic marker Synapsin 1, glutamatergic markers PSD-95 and VGLUT1, GABAergic markers VGAT and Gephyrin, and cholinergic markers VAChT and ChAT. (D) The average cross-correlation between all pairs of raw images of pre- and post-synaptic antibody stains (n = 8 image stacks from 3 *Chat^ires-Cre* mice). (F) The average co-variance between all pairs of raw images specifically within a mask created by the Synaptophysin-YFP stain. Co-variance within YFP terminals is not limited to values between −1 and 1 because all antibody stains were z-scored prior to masking and calculating the co-variance, and therefore antibody signals may be more or less concentrated within the YFP mask (n = 8 image stacks from 3 mice). High covariance specifically between GABAergic and cholinergic proteins emerge when limiting analysis to signal within YFP masks (E), but not the entire image, where correlations between glutamatergic markers are highest (D). (F) Summary of antibody colocalization analysis. First, a 3D mask of the YFP signal is created corresponding to the ChAT⁺ presynaptic terminals. Next, each punctum of antibody signal is assigned a pixel corresponding to where a Gaussian fit of fluorescence has the highest intensity. Then, the colocalization of each antibody pixel within the YFP terminals is determined and a z-score calculated by comparing to the colocalization from 1000 rounds of randomized antibody pixel locations. (G) Colocalization z-scores across antibodies for all samples. Higher positive z-scores indicate relative enrichment of antibody puncta within YFP terminals compared to randomized controls, while negative scores indicate depletion of antibody puncta within YFP terminals (see *Figure 4—figure supplement 2*). Tissue samples from *Chat^ires-Cre* mice are shown, as well as the VGAT antibody z-score from VGAT conditional knock-out mice (6 image stacks from 3 *Chat^ires-Cre* x *Slc32a1^fl/fl* mice), and VAChT antibody z-scores from *Vip^ires-Cre* (5 image stacks from 3 mice) and *Vip^ires-Cre* x *Slc18a3^fl/fl* (6 image stacks from 3 mice). Dashed lines indicate ± 5 z-scores.

The online version of this article includes the following figure supplement(s) for figure 4:

**Figure supplement 1.** Average cross-correlations of synaptic antibody image arrays from *Chat^ires-Cre*, VGAT mosaic knock-out (*Chat^ires-Cre* x *Slc32a1^fl/fl*), *Vip^ires-Cre*, and VAChT mosaic knockout mice (*Vip^ires-Cre* x *Slc18a3^fl/fl*).

**Figure supplement 2.** Summary antibody colocalization with VIP⁺/ChAT⁺ pre-synaptic terminals.

ChAT, and VAChT were consistently enriched within cortical ChAT⁺ interneuron terminals, whereas PSD-95 and VGLUT1 were specifically depleted (*Figure 4G*). Because both VAChT and VGAT expression are central to our conclusions, we validated the specificity of signals from these two antibodies using genetically mosaic conditional knockouts in which the gene encoding each protein was selectively knocked-out from VIP⁺ and ChAT⁺ neurons, respectively (*Vip^ires-Cre* x *Slc18a3^fl/fl,50* and *Chat^ires-Cre* x *Slc32a1^fl/fl*, *Tong et al., 2008*). For each antibody, both its higher covariance with other GABAergic and cholinergic proteins and its enrichment within synaptophysin-YFP-labeled cortical VIP⁺/ChAT⁺ terminals were eliminated when we conditionally deleted VGAT or VAChT (*Figure 4*, *Figure 4—figure supplements 1* and *2*).

Given these data showing that both GABA and ACh release machinery are generally expressed in the pre-synaptic terminals, we examined whether individual terminals and axon segments of VIP⁺/ChAT⁺ neurons differ in their potential to release ACh or GABA. Pre-synaptic terminals of motor cortex VIP⁺/ChAT⁺ neurons were labeled by injection of Cre-dependent synaptophysin-mCherry AAV (AAV(8)-CAG-DIO-synaptophysin-mCherry) into the motor cortex of *Chat^ires-Cre* mice. We classified terminals as GABAergic or cholinergic by antibody staining against VGAT and VAChT, respectively (*Figure 5A*). Compared to array tomography, the thicker slices make it easier to follow individual axons with many putative pre-synaptic terminals. These data show that individual cortical VIP⁺/ChAT⁺ terminals have highly variable expression of VAChT (*Figure 5B–D*), with some axon stretches entirely lacking VAChT (*Figure 5B*), others being entirely positive for VAChT (*Figure 5C*), and some stretches presenting intermingled VAChT-containing and VAChT-lacking terminals (*Figure 5D*).

Quantification of VAChT intensity within VIP⁺/ChAT⁺ individual terminals shows a range of VAChT expression, including strongly labeled terminals and others whose labeling intensities overlap with negative control intensities, which were calculated by measuring the overlap of the pre-synaptic terminal image mask rotated 90 degrees with respect to the VAChT signal image (*Figure 5E*). Overall, VAChT and VGAT intensities positively correlated across terminals ($R^2$ = 0.33, *Figure 5F*), though a population of VGAT-expressing terminals lacking VAChT were found. We categorized each terminal as positive or negative for each vesicular transporter according to a fluorescence intensity threshold that maximally separates VAChT or VGAT signal from the background of each image. By this classification, the majority of terminals are positive for both VGAT and VAChT (*Figure 5G*, $R^2$ = 0.232), with a subset that are positive for VGAT but not VAChT (*Figure 5G*, $R^2$ = 0.091). This held true across a range of classification thresholds. The likelihood that a terminal is VGAT⁺ increases monotonically as the threshold for VAChT is raised, while the proportion of VAChT⁺ terminals plateaus around 75% even at very high thresholds for VGAT (*Figure 5—figure supplement 1A,B*). In other words, strong expression of VAChT ensures co-expression of VGAT, but many highly VGAT-

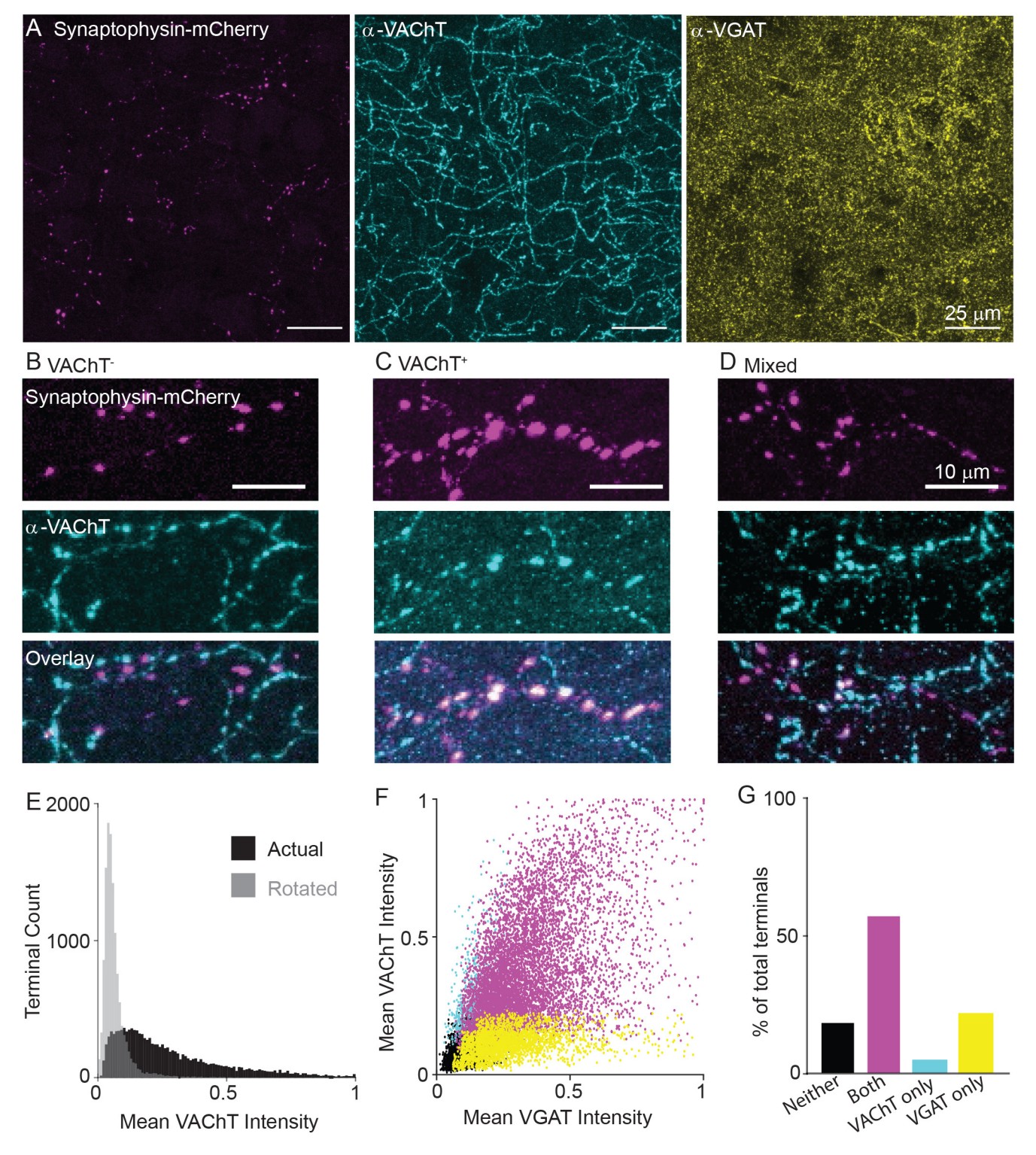

**Figure 5.** Variable expression of VAChT in the presynaptic terminals of cortical ChAT⁺ neurons. (A) Example images of cortical ChAT⁺ pre-synaptic terminals labeled with AAV(8)-DIO-Synaptophysin-mCherry injected in the motor cortex of *Chat^{ires-Cre}* mice. Left: Synaptophysin mCherry; Middle: VAChT immunostain; Right: VGAT immunostain. (B–D) Example images showing putative Synaptophysin-mCherry⁺ axons, with VAChT⁻ terminals (B), VAChT⁺ terminals (C), and intermingled terminals that are both VAChT⁺ and VAChT⁻. (D). (E) Histogram of mean VAChT fluorescence intensity within Synaptophysin-mCherry⁺ terminals. Black histogram represent the actual VAChT intensities, grey histogram represents the mean VAChT intensities when the mCherry image mask is rotated 90˚ relative to the VAChT immunostain image. (F) Scatter plot of mean VGAT intensity and VAChT intensity in

*Figure 5 continued on next page*

*Figure 5 continued*

each putative pre-synaptic terminals (n = 12,356 putative terminals from 30 image stacks from 3 *Chat^{ires-Cre}* mice). Terminals are color-coded according to expression of VAChT and VGAT (Black – neither VGAT or VAChT, Magenta – both VGAT and VAChT, Cyan – VAChT only, Yellow – VGAT only). (**G**) Quantification of the number of terminals of each type in (**F**).

The online version of this article includes the following figure supplement(s) for figure 5:

**Figure supplement 1.** VAChT is present in a subset of terminals across intensity thresholds, and is absent from the terminals of Sst[+] interneurons.

expressing terminals lack VAChT. We also analyzed both VAChT and VGAT expression in pre-synaptic terminals across cortical layers, finding a small but significant decrease in VAChT[+] and increase in VGAT[+] terminals in layer 1 (*Figure 5—figure supplement 1C,D*). As an additional negative control, we repeated this analysis in *Sst^{ires-Cre}* mice, and confirmed that terminals of Sst[+] interneurons, which do not express *Slc18a3*, were almost completely negative for VAChT protein, with no relationship between VGAT and VAChT fluorescence intensity per terminal (*Figure 5—figure supplement 1E–I*). In summary, we identified two different populations of terminals, those capable of releasing both GABA and ACh and those capable of releasing only GABA. This suggests that release of ACh from these neurons is likely to be targeted to specific post-synaptic neurons.

## A sparse population of non-VIP ChAT+ neurons specific to the mPFC

A recent publication reported rates of GABA and ACh connectivity from cortical ChAT[+] neurons that were strikingly different from what we described above (*Obermayer et al., 2019*). They reported that optogenetic activation of cortical ChAT[+] neurons frequently resulted in postsynaptic cholinergic currents and rarely GABAergic currents. To reconcile these results with ours, we compared the experimental conditions in the two studies. In addition to differences in the composition of recording solution, a major difference between our studies is the choice of brain regions - their connectivity analysis was restricted to medial prefrontal cortex (mPFC), whereas the majority of our experiments were conducted in motor cortex (M1).

We first compared connectivity to layer 1 interneurons between mPFC and M1 from all forebrain cholinergic neurons using a mouse line that expressed ChR2 in all cholinergic neurons (*Chat^{ires-Cre}* x *Rosa26^{lsl-ChR2-EYFP}*, *Figure 6A*). To our surprise, we found significant differences in the proportion of cholinergic responses between M1 and mPFC, with more frequent cholinergic responses in the latter (*Figure 6B–D*), indicating a fundamental difference in cholinergic innervation of these two cortical regions. To determine if these differences could be explained by ACh release from local cortical cholinergic interneurons, we injected AAV-encoding Cre-dependent ChR2-mCherry directly to the mPFC and M1 in *Chat^{ires-Cre}* mice and compared synaptic responses in layer 1 interneurons across brain regions (*Figure 6E*). We found a reduced rate of overall connectivity, most likely due to lack of ChR2 expression in basal forebrain projections to cortex and incomplete transduction of cortical ChAT[+] neurons with AAV. Nevertheless, we observed that a significantly larger proportion of layer 1 neurons receives cholinergic input in the mPFC compared to in M1 (20/131 neurons with nAChR responses in mPFC compared to 1/43 in M1) and significantly fewer proportion of GABAergic responses (9/131 neurons with GABA_AR responses compared to 13/43 in M1, *Figure 6F–H*). These results could not be explained by other major differences between our studies such as the brain slice cutting solution (*Figure 6—figure supplement 1A,B*) and the internal whole-cell recording solution (*Figure 6—figure supplement 1A,C*). Indeed, within the mPFC our results are consistent with those of Obermayer et al, and indicate a difference in the connectivity of local cholinergic neurons between the mPFC and motor cortex.

To test whether this difference in connectivity across brain regions is specific to VIP[+]/ChAT[+] neurons and to eliminate the possibility of contamination from long-range cholinergic axons, we repeated this experiment using mice that express ChR2 in all VIP[+] interneurons (*Vip^{ires-Cre}* x *Rosa26^{lsl-ChR2-EYFP}*). We reasoned that because sub-cortical cholinergic neurons do not express *Vip* (*Figure 6—figure supplement 2A–C*), any cholinergic responses elicited by optogenetic stimulation of VIP[+] interneurons would be attributable to local cortical VIP[+]/ChAT[+] neurons (*Figure 6I*). However, we only identified a single cholinergic response in mPFC, with the majority of synaptic responses from VIP[+] interneuron activation in mPFC and M1 being GABAergic (*Figure 6J–L*). In additional recordings, we included gabazine to block GABA_AR-mediated responses and allow for

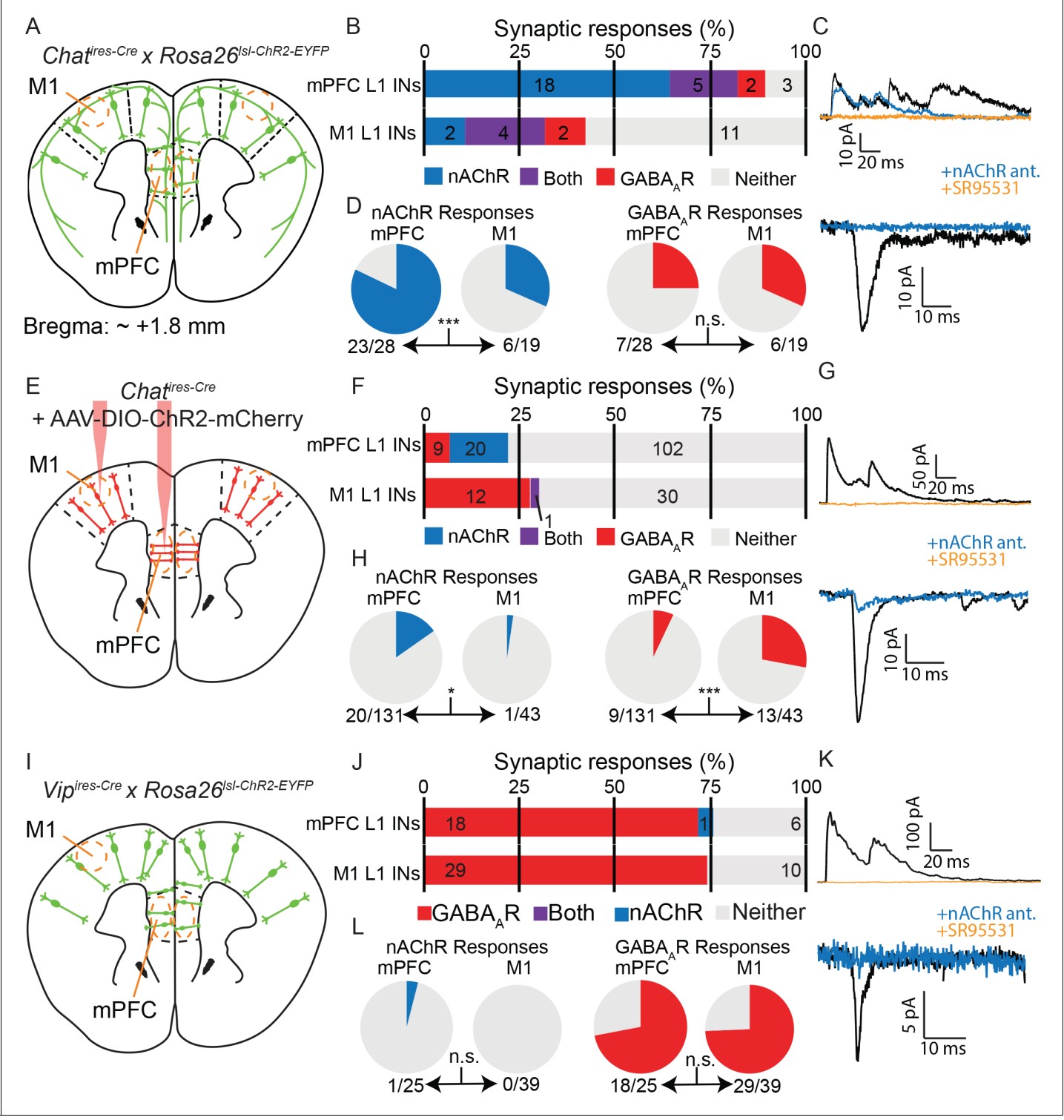

**Figure 6.** Cholinergic connectivity to Layer 1 interneurons is greater in medial pre-frontal cortex than motor cortex, but not from VIP[+] interneurons. (**A**) Experimental design: Acute coronal slices taken ~1.8 mm anterior to bregma were obtained from mice where all cholinergic neurons express ChR2 (*Chat^{ires-Cre}* x *Rosa26^{lsl-ChR2}*) and whole-cell voltage clamp recordings made from layer 1 interneurons in mPFC and M1. (**B**) Proportion of layer 1 interneurons showing nAChR-mediated and GABA_AR-mediated synaptic responses, as determined by clamping the cell to −70 mV and 0 mV, respectively. (n = 47 total neurons from 5 *Chat^{ires-Cre}* x *Rosa26^{lsl-ChR2}* mice). (**C**) Example traces of GABA_AR-mediated (top) and nAChR-mediated synaptic responses (bottom), as confirmed by block by gabazine and a cocktail of nAChR antagonists (DHβE, MLA, MEC). (**D**) Direct comparison of the proportion of cells showing nAChR-mediated responses (left, p=0.0007) or GABA_AR-mediated responses (right, p=0.7431). (**E–H**) Same as (**A–D**), but

*Figure 6 continued on next page*

*Figure 6 continued*

only cortical ChAT$^+$ neurons now express ChR2, through injection of Cre-dependent ChR2 virus to both M1 and mPFC (AAV(8)-DIO-ChR2-mCherry). Direct comparison of the proportion of cells with nAChR-mediated responses (bottom) shows a significant decrease in M1 compared to mPFC (left, p=0.0301), while those with GABA$_A$R-mediated responses show a significant increase (right, p=0.0002; n = 174 total neurons from 9 *Chat*$^{ires-Cre}$ mice). (I–L) Same as (A–C) and (E–G), but for acute coronal slices with all VIP$^+$ interneurons expressing ChR2 (*Vip*$^{ires-Cre}$ x *Rosa26*$^{lsl-ChR2-EYFP}$). (n = 64 neurons from 7 *Vip*$^{ires-Cre}$ x *Rosa26*$^{lsl-CHR2-EYFP}$ mice). Example of the sole nAChR-mediated synaptic response obtained from a Layer 1 interneuron following optogenetic stimulation of VIP neurons is shown in (K). Direct comparison of the proportion of cells with nAChR- or GABA$_A$R-mediated responses is not significantly different between mPFC and M1 (p=0.3906 and p=1, respectively). Single asterisk (*) indicates significance at p<0.05, triple asterisk (***) indicates significance at p<0.001 and all p-values calculated by Fisher's exact test.

The online version of this article includes the following figure supplement(s) for figure 6:

**Figure supplement 1.** Difference in ChAT$^+$ interneuron connectivity are not explained by differences in slicing solution or whole-cell internal recording solution.

**Figure supplement 2.** Synaptic connectivity from VIP$^+$ neurons, which excludes basal forebrain cholinergic neurons, results in little to no detectable muscarinic ACh receptor-mediated responses.

**Figure supplement 3.** ACh-mediated synaptic connectivity is not effected by circadian cycle or viral injection.

more rapid screening of nAChR-mediated responses, but did not find additional nAChR-mediated responses (*Figure 6—figure supplement 2D,E*). We also screened for potential muscarinic ACh receptor responses from VIP$^+$ interneurons throughout the cortex, with and without acetylcholine esterase inhibition to increase the size and duration of potential responses, but did not identify synaptic responses that we could confirm to be mediated by release of acetylcholine (*Figure 6—figure supplement 2F–J*). We tested the possibilities that VIP$^+$ interneurons could be induced to be cholinergic in mice raised in a reverse light cycle (*Figure 6—figure supplement 3A,B*), in case the circadian cycle, which also differed between our study and that of Obermayer et al, caused changes in the ability of VIP$^+$ interneurons to release ACh. We also tested exposure to isoflurane (*Figure 6—figure supplement 3A,C*), and viral delivery of ChR2 (*Figure 6—figure supplement 3D,E*), to rule out the possibility that the act of delivering virus, which is unnecessary when surveying connectivity from all VIP$^+$ interneurons, induced a switch to more cholinergic signaling. However, none of these manipulations increased the rate of cholinergic synaptic responses. This relative lack of cholinergic responses from the wider population of VIP$^+$ interneurons is more similar to the connectivity observed from cortical VIP$^+$/ChAT$^+$ neurons in the motor cortex.

If not arising from VIP-expressing ChAT$^+$ interneurons, what is the local source of cholinergic inputs to layer 1 interneurons in mPFC? When performing our connectivity analysis in the mPFC, we noticed the presence of ChR2-EYFP$^+$ neurons with strikingly different morphology than typical VIP$^+$/ChAT$^+$ interneurons, with larger cell bodies and an orientation parallel to the cortical surface (*Figure 7A*) instead of perpendicular (*Figure 1*). Fluorescent in situ hybridization for *Chat* and *Vip* revealed a sparse population of neurons in the mPFC that express high levels of *Chat* but not *Vip* mRNA (*Figure 7B*).

Because these neurons lacked *Vip* expression, we hypothesized that they derive from a different developmental origin than the VIP$^+$/ChAT$^+$ interneurons. VIP interneurons are derived from the caudal ganglionic eminence, whereas most sub-cortical cholinergic neurons develop from medial ganglionic eminence progenitors that are marked by transient expression of the transcription factor Nkx2.1 (*Magno et al., 2017*; *Allaway and Machold, 2017*; *Figure 7C*). Indeed, using an intersectional genetic strategy (*Plummer et al., 2015*; *He et al., 2016*) to label neurons that express, even transiently, both *Chat* and *Nkx2.1* (*Chat*$^{ires-Cre}$ x *Nkx2.1*$^{ires-Flp}$ x RC:FLTG), we identified Nkx2.1-lineage neurons in the mPFC that immunolabel for ChAT but not for VIP (*Figure 7C,D*). These neurons were exceptionally sparse - both non-VIP, ChAT$^+$ neurons identified by FISH (*Figure 7B*) and Nkx2.1$^+$/ChAT$^+$ neurons identified genetically (*Figure 7C,D*), only 3–5 neurons were identifiable in each analyzed mouse brain, indicating that these are the same population of cells (see additional examples of morphology and orientation in *Figure 7—figure supplement 1*). We did not find any examples of genetically-labeled Nkx2.1$^+$/ChAT$^+$ neurons in other regions of the cortex in a whole-brain survey, suggesting that the low rate of non-VIP, ChAT$^+$ neurons we report in *Figure 2A & B* is due to false negatives as a result of incomplete labeling and does not indicate the presence of Nkx2.1$^+$/ChAT$^+$ neurons outside of the mPFC. Given the existence of these non-VIP, ChAT$^+$ neurons in the mPFC, we repeated our connectivity analysis as described above by injecting AAV-encoding

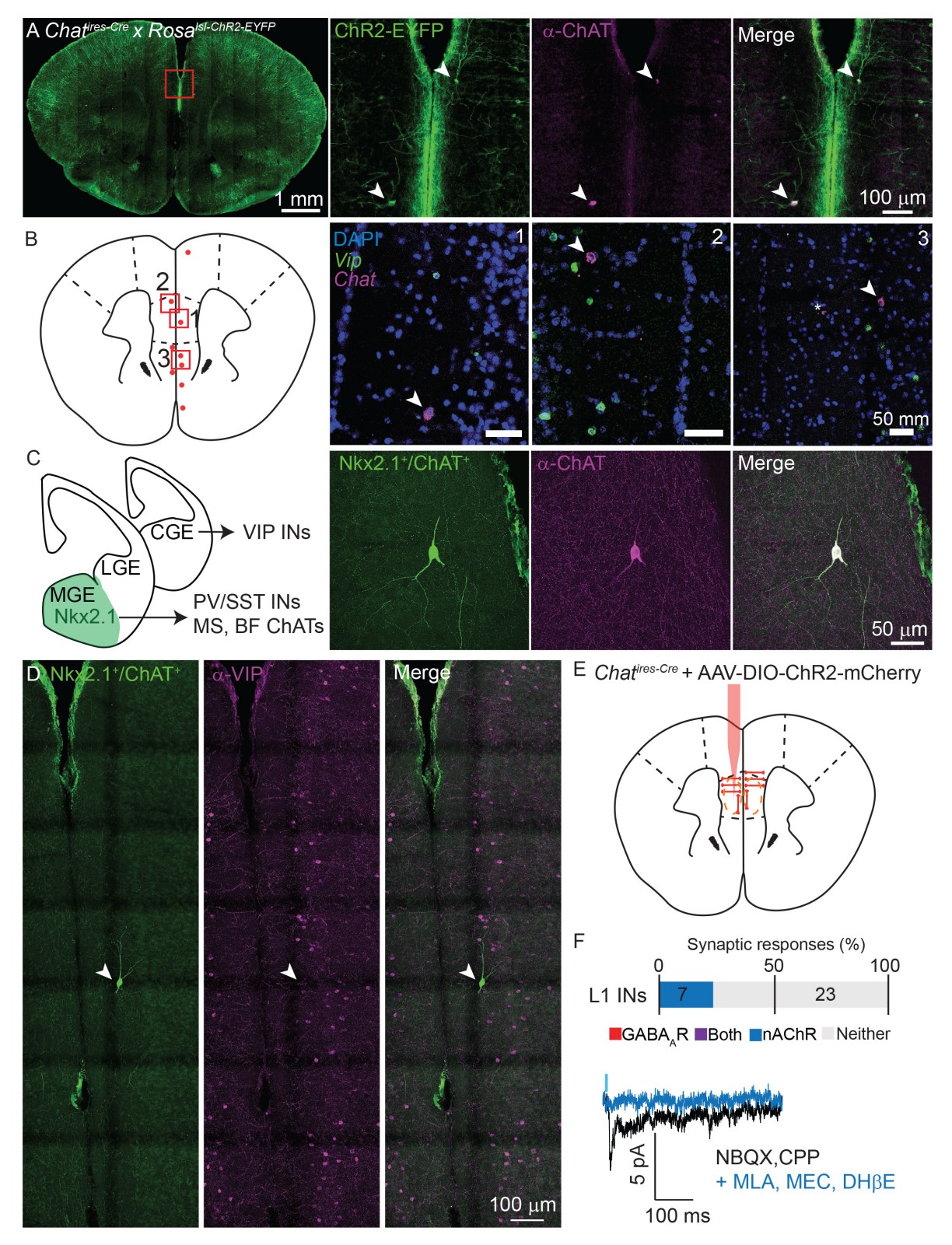

**Figure 7.** Non-VIP ChAT⁺ neurons contribute to cholinergic connectivity in the mPFC. (**A**) Example images from a *Chat*^ires-Cre x *Rosa26*^lsl-CHR2-EYFP mouse. In the mPFC (red inset, left panel), two neurons are shown that are oriented parallel to the cortical surface, as opposed to the typical perpendicular orientation, (middle left panel, arrowheads) which have strong staining against ChAT (middle right panel, arrowheads). (**B**). Fluorescent in situ hybridization from mPFC reveals a sparse population of cells with strong labeling for *Chat*, but not for *Vip*. Approximate location of *Chat*-

*Figure 7 continued on next page*

*Figure 7 continued*

expressing, *Vip*-lacking neurons across 3 different mice are indicated by red dots, and the locations of three example images (right panels) shown by red boxes. Arrowheads indicate *Vip*-lacking, *Chat*$^+$ neurons, the asterisk in panel 3 indicates a nearby neuron expressing both *Vip* and *Chat*. (**C**) Cortical VIP$^+$ neurons develop from the caudal ganglionic eminence (CGE), whereas PV$^+$, Sst$^+$, and most cholinergic neurons of the medial septum (MS) and basal forebrain (BF), derive from Nkx2.1-expressing neurons of the medial ganglionic eminence (MGE). Neurons genetically labeled by transient expression of Nkx2.1 and ChAT (*Chat*$^{ires-Cre}$ x *Nkx2.1*$^{ires-FLP}$ x RC::FLTG, middle left panel) exist in the mPFC that strongly label for ChAT protein (middle right panel) in adult mice. (**D**) Example Nkx2.1$^+$/ChAT$^+$ neuron (left panel) demonstrating a lack of colabeling with *Vip* (middle, right panels). (**E**) Experimental design: AAV(8)-DIO-ChR2-mCherry is injected directly into the mPFC and allowed to express for 3 weeks. Whole-cell voltage clamp recordings were then taken from layer 1 interneurons in the immediate vicinity of putative non-VIP ChAT$^+$ neurons, as indicated by their morphology and large soma compared to VIP$^+$ interneurons. (**F**) Summary of the proportion of layer 1 interneurons showing synaptic responses following optogenetic stimulation of nearby putative Non-VIP ChAT$^+$ neurons. GABA$_A$R- and nAChR-mediated synaptic responses are differentiated based on reversal potential and sensitivity to selective antagonists. Example nAChR-mediated synaptic response recorded at $-70$ mV in the presence of glutamatergic antagonists NBQX and CPP, blocked by nAChR antagonists DH$\beta$E, MLA, and MEC. No GABA$_A$R-mediated synaptic responses were observed near putative Non-VIP ChAT$^+$ neurons in the absence of other nearby VIP$^+$/ChAT$^+$ neurons. (n = 30 neurons from 6 *Chat*$^{ires-Cre}$ mice).

The online version of this article includes the following figure supplement(s) for figure 7:

**Figure supplement 1.** Example images of Non-VIP, ChAT$^+$/Nkx2.1$^+$ neurons in the mPFC.

Cre-dependent ChR2-mCherry into the mPFC of *Chat*$^{ires-Cre}$ mice. We focused only on potential post-synaptic layer 1 interneurons in the immediate vicinity of ChR2-expressing non-VIP ChAT$^+$ neurons, identifiable by their unique morphology and relatively large soma. Indeed, layer 1 interneurons near ChR2-expressing non-VIP ChAT$^+$ neurons (but without nearby VIP$^+$/ChAT$^+$ neurons) received nicotinic cholinergic following optogenetic stimulation, but not GABAergic synaptic currents (*Figure 7E*). Thus, we have identified a previously unknown MGE-derived population of non-VIP cholinergic neurons that explain the differential synaptic connectivity of local ChAT$^+$ neurons between mPFC and other regions of cortex. Given that only a few of these neurons are present in any given mouse brain, and because they are not present outside of the mPFC, their absence from systematic surveys of cortical cell classes is not surprising.

## Discussion

In this study, we characterized the synaptic physiology and anatomy of cortical ChAT$^+$ interneurons, focusing primarily on the vast majority that are a subset of VIP$^+$ interneurons. These VIP$^+$/ChAT$^+$ interneurons co-transmit both GABA and ACh, targeting each neurotransmitter onto different post-synaptic neurons. ACh transmission is sparse and primarily targets layer 1 interneurons and other VIP$^+$/ChAT$^+$ interneurons. In contrast, GABA transmission is widespread onto inhibitory interneuron subtypes, especially Sst$^+$ interneurons, a pattern of GABAergic connectivity that is consistent with previous analyses of VIP$^+$ cells (*Pfeffer et al., 2013*; *Karnani et al., 2016a*). These functional results are complemented by imaging data showing pre-synaptic specialization of the ability of VIP$^+$/ChAT$^+$ neurons to release ACh and GABA. However, given the enrichment for ACh synthesis and release proteins in the pre-synaptic terminals and our finding that the majority of VIP$^+$/ChAT$^+$ synapses are competent to release both ACh and GABA, the relative sparsity of ACh-mediated responses remains surprising.

Given the low number of post-synaptic cholinergic responses we observed, there are several possibilities for the function and synaptic logic of ACh release. One possibility is that these neurons release ACh onto a distinct post-synaptic target we have not identified molecularly. Although we attempted to be as comprehensive as possible in surveying potential post-synaptic targets, the full diversity of cortical cellular subtypes is only beginning to be understood, and it is therefore possible that future research will identify a neuronal subtype that is strongly innervated by ACh released from VIP$^+$/ChAT$^+$ synapses. Similarly, ACh released by VIP$^+$/ChAT$^+$ interneurons may have primarily metabotropic effects, including generative post-synaptic effects that we are unable to detect electrophysiologically. We searched extensively for post-synaptic responses mediated by muscarinic ACh receptors without success (*Figure 6—figure supplement 2D–H*), but we did not examine potential changes in cellular excitability or synaptic plasticity. Another possibility is that ACh transmission is indeed very sparse, but highly targeted, activating specific subnetworks of disinhibitory interneurons. This possibility is consistent with previous studies showing that the firing of VIP$^+$ interneurons can

recruit other VIP[+] interneurons via nAChRs (*Karnani et al., 2016b*). Finally, ACh transmission from VIP[+/]ChAT[+] neurons may be conditional such that it only occurs in certain contexts or developmental epochs. Precedent for this has been observed following critical period plasticity in the visual and auditory cortex, when increased expression of Lynx1 blocks nAChR-mediated signaling to limit synaptic plasticity, and deletion of Lynx1 can reveal previously masked nAChR-mediated currents (*Morishita et al., 2010*; *Takesian et al., 2018*) Regulation of neurotransmitter release has also been observed in other systems, such as the retina where ACh and GABA are differentially released by starburst amacrine cells depending on the direction of the light stimulus (*Lee et al., 2010a*; *Sethuramanujam et al., 2016*), and in several examples where neurons appear to switch their predominant neurotransmitter during development or after bouts of neuronal activity (*Spitzer, 2015*).

Nevertheless, the pattern of GABA and ACh connectivity we describe suggests a coherent model of the net effect that these neurons have on cortical circuits: ACh-mediated excitation of other layer 1 and VIP[+]/ChAT[+] disinhibitory interneurons, combined with the GABA-mediated disinhibition, provides a powerful activating signal to local cortical areas. This is consistent with previous findings showing that VIP[+] interneurons can promote the cooperative firing of other VIP[+] interneurons partially through activation of nAChRs (*Karnani et al., 2016b*). In different behavioral paradigms, activation of VIP[+] and layer 1 interneurons has been shown to increase the gain of sensory responses in pyramidal neurons (*Pi et al., 2013*; *Fu et al., 2014*) or signal a cue for fear conditioning (*Letzkus et al., 2011*). In each of these cases, these disinhibitory neurons are activated by ascending cholinergic inputs from basal forebrain. We propose that ACh release reinforces and amplifies the cortical activation achieved by the broader VIP[+] interneuron population, itself activated by ascending cholinergic projections, in order to enhance the response to salient cues. Cortical ChAT[+] neurons in barrel cortex are strongly activated by sensory stimuli, supporting this model, but optogenetic activation of cortical ChAT[+] neurons actually slightly decreases the response of other neurons to whisker deflections (*Dudai et al., 2020*), suggesting additional complexities. One explanation is that strong optogenetic activation may occlude the time-locked, stimulus-evoked firing of cortical ChAT[+] neurons, decreasing their ability to boost neuronal activity. Further experiments are necessary to clarify the in vivo functional role of cortical ChAT[+] neurons on cortical processing.

The synaptic connectivity of VIP[+]/ChAT[+] interneurons illustrates several notable modes of synaptic transmission. First, they are a local source of ACh that is sparse and highly targeted, contrasting with the broadly dispersed, long-range projections from the basal forebrain. This raises the possibility of neuromodulation by ACh that occurs within highly specific subnetworks of cortical neurons, as opposed to a bulk signal that affects large regions of cortex at once. This is consistent with a recent reevaluation of cortical ACh signaling as not only a diffuse, tonic signal that operates on relatively slow time scales, but also as a phasic signal that operates on the time scale of seconds and milliseconds (*Sarter et al., 2014*).

Second, they provide an example of specialized pre-synaptic terminals that allow for targeting of different neurotransmitters to specific outputs. Such output-specific targeting of neurotransmitter release is a largely unexplored aspect of synaptic transmission in the brain. Our data are consistent with two different models of output-specific targeting – either multiple subtypes of VIP[+]/ChAT[+] interneurons differentially release ACh, or individual VIP[+]/ChAT[+] interneurons with specialized presynaptic terminals that target different neurotransmitters to specific outputs. Given our observation of individual stretches of axon with intermingled VAChT[+] and VAChT[-] terminals (*Figure 5D*, we favor the model of individual VIP[+]/ChAT[+] neurons that tailor their neurotransmitter output based on the target neuron. A similar level of regulation can be observed in dopaminergic neurons which spatially segregate co-release of glutamate and dopamine in different brain regions (*Stuber et al., 2010*; *Mingote et al., 2015*), and whose individual axons segregate terminals that release dopamine or glutamate (*Zhang et al., 2015*). Similar differentiation of neurotransmitter release has been reported elsewhere in the cholinergic system, specifically in Globus Pallidus externus projections to the cortex (*Saunders et al., 2015b*) and in hippocampus-projecting septal cholinergic neurons that release ACh and GABA from different synaptic vesicles (*Takács et al., 2018*). The possibility for separable release of multiple neurotransmitters adds another level of complexity to our understanding of how neurons communicate.

Third, our study identifies a multiple levels of heterogeneity within cortical ChAT[+] neurons. The first is existence of multiple subtypes of cortical ChAT[+] neurons that differ in developmental origins, molecular profiles, and synaptic connectivity. Whereas the vast majority of cortical ChAT[+] neurons

are a subset of VIP+ interneurons, which are CGE-derived, we discovered a small population (only several neurons present in each mouse brain) of non-VIP ChAT+ neurons derived from Nkx2.1-expressing (MGE and pre-optic area derived) progenitors that are developmentally more similar to cholinergic neurons of the basal forebrain (*Magno et al., 2017*; *Allaway and Machold, 2017*). Notably, within cortex these non-VIP ChAT+ neurons are found only mPFC, potentially why they are not identifiable in recently published single-cell RNA sequencing data sets (*Saunders et al., 2018*; *Tasic et al., 2016*; *Zeisel et al., 2015*) and have evaded widespread notice.

The presence of non-VIP ChAT+ neurons explains the differences in connectivity between our study and a recent one that examind cortical ChAT+ interneuron connectivity in the mPFCanderroneously attributed all local cholinergic connectivity to VIP+/ChAT+ neurons (*Obermayer et al., 2019*). Another major difference between our two studies is that our experiments were conducted entirely in mice, whereas they made extensive use of ChAT-Cre transgenic rats (although reached the same conclusions from their more limited analysis of mice). Rats may have a greater proportion of ChAT+, VIP+ interneurons, raising the possibility that ACh transmission is more prevalent from cortical VIP+/ChAT+ neurons in rat than in mice.

We find that further heterogeneity exists within VIP+/ChAT+ interneurons and their pre-synaptic terminals. We observed two separate terminal populations, one capable of releasing both ACh and GABA, and another capable of releasing only GABA. The presence of stretches of axons with only one class of presynaptic bouton suggests that this heterogeneity exists within axons of single VIP+/ChAT+ neurons or that there are two subpopulations of VIP+/ChAT+ neurons. In either case, individual VIP+/ChAT+ neurons have specialized pre-synaptic terminals that differentially target GABA and ACh onto post-synaptic neurons.

The existence of cortical ChAT+ neurons requires a reevaluation of studies that globally manipulate cholinergic signaling in cortex. While many studies specifically targeted cortically-projecting basal forebrain neurons, several have used genetic crosses that affect all cholinergic neurons in the brain (*Chen et al., 2015*; *Sparks et al., 2017*; *Kuchibhotla et al., 2017*; *Dasgupta et al., 2018*), and therefore include confounding effects from local VIP+/ChAT+ interneurons. Studies that use ChAT-BAC-ChR2 mice to activate cholinergic neurons not only run the risk of confounding gain-of-function effects due to overexpressed VAChT (*Kolisnyk et al., 2013*), but also from incidental manipulation of cortical VIP+ interneurons, which are known to have profound effects on cortical function even purely through GABA release. Going forward, studies of cholinergic signaling in cortex must differentiate between contributions from basal forebrain projections and those from local cholinergic interneurons.

# Materials and methods

**Key resources table**

| Reagent type (species) or resource | Designation | Source or reference | Identifiers | Additional information |
|---|---|---|---|---|
| Strain, strain background (*Mus musculus*) | Wild-type | Jackson Labs | C57BL6/J | Stock#00664 |
| Strain, strain background (*Mus musculus*) | *Chat*$^{ires-Cre}$ | Jackson Labs | B6;129S6-*Chat*$^{tm1(cre)Lowl}$/J | Stock # 006410 |
| Strain, strain background (*Mus musculus*) | *Vip*$^{ires-Cre}$ | Jackson Labs | VIP$^{tm1(cre)Zjh}$/J | Stock # 010908 |
| Strain, strain background (*Mus musculus*) | *Sst*$^{ires-Flp}$ | Jackson Labs | Sst$^{tm3.1(flpo)Zjh}$/J | Stock # 028579 |
| Strain, strain background (*Mus musculus*) | *Rosa26*$^{lsl-tdTomato}$, (Ai14) | Jackson Labs | B6.129Sg-Gt(ROSA)26Sor$^{tm14(CAG-tdTomato)Hze}$/J | Stock # 007908 |

*Continued on next page*

*Continued*

| Reagent type (species) or resource | Designation | Source or reference | Identifiers | Additional information |
|---|---|---|---|---|
| Strain, strain background (*Mus musculus*) | *Rosa26*$^{lsl-ChR2-EYFP}$, (Ai32) | Jackson Labs | B6;129S-Gt(ROSA) 26Sor$^{tm32(CAG-COP4*H134R/EYFP)Hze}$/J | Stock # 012569 |
| Strain, strain background (*Mus musculus*) | Sst-GFP (GIN) | Jackson Labs | FVB-Tg (GadGFP) 45704Swn/J | Stock # 003718, referred to as "GIN" mice by the Jackson Laboratory, as "Sst-GFP" mice in the text. |
| Strain, strain background (*Mus musculus*) | PV-GFP (G42) | Jackson Labs | CB6-Tg(GAD1-EGFP)G42zjh/J | Stock # 007677, referred to as "GAD67-GFP" or "G42 line" mice by the Jackson laboratory, as "PV-GFP" mice in the text. |
| Strain, strain background (*Mus musculus*) | 5HT3aR-GFP | Gift from B. Rudy lab (NYU) | Tg(Htr3a-EGFP) DH30Gsat | |
| Strain, strain background (*Mus musculus*) | *Gad2*$^{ires-GFP}$ | Jackson Labs | Gad2$^{tm2(cre)Zjh}$/J | Stock # 010802 |
| Strain, strain background (*Mus musculus*) | *Slc18a3*$^{fl/fl}$ | Gift of V. and M. Prado (UWO) | VAChT$^{flox/flox}$ | *Martins-Silva et al., 2011* |
| Strain, strain background (*Mus musculus*) | *Slc32a1*$^{fl/fl}$ | Jackson Labs | Slc32a1$^{tm1Lowl}$/J | Stock # 012897 |
| Strain, strain background (*Mus musculus*) | *Nkx2.1*$^{ires-Flp}$ | Jackson Labs | Nkx2-1$^{tm2.1(flop)Zjh}$ | Stock # 028577 |
| Strain, strain background (*Mus musculus*) | RC::FLTG | Jackson Labs | B6.Cg-Gt(ROSA)26 Sor$^{Tm1.3(CAG-tdTomato,-EGFP)Pjen}$/J | Stock #026932 |
| Strain, strain background (AAV) | AAV(8)-EF1α-hChR2(H134R)-mCherry-WPRE-pA | UNC Vector Core | | Titer:~2×10$^{13}$ gc/ml |
| Strain, strain background (AAV) | AAV(DJ)-EF1α-fDIO-EYFP | BCH Viral Core | | Titer:~2×10$^{12}$ gc/ml |
| Strain, strain background (AAV) | AAV(1)- EF1α-fDIO-ChR2-EYFP | BCH Viral Core | | Titer:~5×10$^{13}$ gc/ml |
| Strain, strain background (AAV) | AAV(8)- EF1α-DIO-FlpO | BCH Viral Core | | Titer:~7×10$^{11}$ gc/ml |
| Strain, strain background (AAV) | AAV(8)- EF1α-DIO-mCherry | UNC Vector Core | | Titer:~6×10$^{12}$ gc/ml |
| Strain, strain background (AAV) | AAV(8)-CMV-DIO-Synaptophysin-EYFP | UNC Vector Core | | Titer:~6×10$^{12}$ gc/ml |
| Strain, strain background (AAV) | AAV(9)-CAG-DIO-Synaptophysin-mCherry | MIT McGovern Viral Core | | Titer:~2×10$^{13}$ gc/ml |
| Antibody | chicken α-GFP | GeneTex | GTC13970 | Dilution: 1:500 |
| Antibody | mouse α-Gephyrin | Biosciences Parmingen | 612632 | Dilution: 1:500 |
| Antibody | rabbit α-Synapsin-1 | Cell Signaling Technology | 5297S | Dilution: 1:500 |

*Continued on next page*

*Continued*

| Reagent type (species) or resource | Designation | Source or reference | Identifiers | Additional information |
|---|---|---|---|---|
| Antibody | rabbit α-PSD95 | Cell Signaling Technology | 3450 | Dilution: 1:500 |
| Antibody | rabbit α-VGAT | Synaptic Systems | 131 011 | Dilution: 1:500 |
| Antibody | mouse α-VAChT | Synaptic Systems | 139 103 | Dilution: 1:500 |
| Antibody | guinea pig α-VGAT | Millipore | AB5905 | Dilution: 1:500 |
| Antibody | goat α-ChAT | Millipore | AB144P | Dilution: 1:500; |
| Antibody | Rabbit α-VIP | ImmunoStar | 20077 | Dilution: 1:500 |
| Antibody | Rabbit α-Somatostatin, Clone YC7 | Millipore | MAB354 | Dilution: 1:500 |
| Antibody | mouse α-Parvalbumin | Millipore | MAB1572 | Dilution: 1:500 |
| Sequence-based reagent | Mm-Chat-C2 | ACDBio | 408731-C2 | |
| Sequence-based reagent | Mm-Slc32a1 | ACDBio | 319191 | |
| Sequence-based reagent | Mm-Gad1-C3 | ACDBio | 400951-C3 | |
| Sequence-based reagent | Mm-Gad2-C3 | ACDBio | 439371-C3 | |
| Sequence-based reagent | Mm-Slc5a7-C3 | ACDBio | 439941-C3 | |
| Sequence-based reagent | Mm-Slc18a3-C3 | ACDBio | 448771-C3 | |
| Sequence-based reagent | Mm-Vip | ACDBio | 4159341 | |
| Sequence-based reagent | Cre-01-C3 | ACDBio | 474001-C3 | |
| Commercial assay or kit | RNAscope Fluorescent Multiplex Detection Reagents | ACDBIO | 320851 | |

## Mice

All mice used in this study were between 2 and 4 months in age. For experiments using only *Chat*$^{ires-Cre}$ mice, homozygous mice were maintained. For all crosses of two or more mouse lines, homozygous breeders were used to produce heterozygous off-spring for experiments, with the exception of experiments requiring conditional deletion of VGAT or VAChT, in which case homozygous *Slc32a1*$^{fl/fl}$ or *Slc18a3*$^{fl/fl}$ conditional knock-out mice were produced that were either homozygous or heterozygous for *Chat*$^{ires-Cre}$ or *Vip*$^{ires-Cre}$, respectively. All mice were maintained in a 12 hr light-dark cycle, with the light cycle occurring between 7 am and 7 pm, with the exception of a cohort of mice in *Figure 6—figure supplement 3*, which lived in a reverse 12 hr light cycle. All experiments were performed according to animal care and use protocols approved by the Harvard Standing

Committee on Animal Care in compliance with guidelines set for in the NIH *Guide for the Care and Use of Laboratory Animals*.

## Virus injections

For intracranial injection of virus, the surgery work area was maintained in aseptic conditions. Mice were anesthetized with 2–3% isoflurane and given 5 mg/kg ketoprofen as prophylactic analgesic, and placed on a heating pad in a stereotaxic frame (David Kopf Instruments) with continuous delivery and monitoring of appropriate isoflurane anesthesia. For one set of experiments (*Figure 6—figure supplement 3*), a cohort of mice were put under isoflurane anesthesia for two hours and allowed to recover without subsequent surgery. For the surgery, the skin above the skull was carefully cleared of hair with scissors and depilatory cream (Nair) and sterilized with alternating scrubs with alcohol pads and betadine pads. A midline incision was made in the skin and the skull exposed and small holes drilled into the skull at the appropriate coordinates depending on the injection site. For injections into motor cortex, injection coordinates were (relative to bregma):±1.8 mm ML, + 1.8 mm and + 0.5 mm AP, and −0.6 mm from the pia. Visual cortex was targeted by injecting (from lambda):±2.5 mm ML, 0 mm AP, −0.25 mm from the pia. 200–500 nl of the appropriate virus was injected through a pulled glass pipette at a rate of 100 nl/min with a UMP3 microsyringe pump (World Precision Instruments) for each of these injection sites. For targeting medial prefrontal cortex, injection coordinates were (from bregma):±0.4 mm ML, + 1.8 mm AP, and −2.0 and −1.3 mm from pia. 250 nl of virus was injected at each of the mPFC sites as above. Following injection, the pipette was allowed to sit for 10 min to prevent leak of the virus from the injection site, and then the glass pipette slowly removed over the course of 1–2 min. Following surgery, mice were monitored in their home cage for 4 days following surgery, and received daily analgesia for 2 days following surgery. Mice were sacrificed for experiments at least 3 weeks following injection to allow for robust viral expression. When we injected multiple viruses, they were mixed in equal proportions.

## Electrophysiology

300 μm acute coronal brain slices were prepared from mice deeply anesthetized with isoflurane inhalation and perfused with ice-cold cutting solution containing (in mM): 25 NaHCO$_3$, 25 Glucose, 1.25 NaH$_2$PO$_4$, 7 MgCl$_2$, 2.5 KCl, 0.5 CaCl2, 11.6 ascorbic acid, 3.1 pyruvic acid, 110 Choline chloride. Following dissection, brains were blocked by cutting along the mid-sagittal axis, and brains glued to the platform along the mid-sagittal surface before slicing on a Leica VT1000s vibratome, while maintaining submersion in cold choline cut solution. Following cutting, slices recovered for 30–45 min in 34° C artificial cerebral spinal fluid (aCSF) containing (in mM): 125 NaCl, 2.5 KCl, 1.25 NaH$_2$PO$_4$, 25 NaHCO$_3$, 11 glucose, 2 CaCl$_2$, 1 MgCl$_2$. Subsequently all recording took place in continuous perfusion (2–3 ml/min) of room temperature aCSF. Both the choline cut solution and aCSF were continuously equilibrated by bubbling with 95%–0$_2$/5% CO$_2$.

After recovery, slices were transferred to a recording chamber mounted on an upright microscope (Olympus BX51WI). Cells were imaged using infrared-differential interference contrast with a 40x water-immersion Olympus objective. To confirm ChR2 expression and GFP-labeled interneurons, we used epifluorescence with an X-Cite 120Q (Excelitas) as a light source. Whole cell voltage-clamp and current clamp recordings were obtained by forming intracellular seals with target neurons with patch pipettes pulled from borosilicate glass (BF150-86-7.5, Sutter). Pipettes (2–4 MOhm pipette resistance) were pulled with a P-97 flaming micropipette puller (Sutter). Pipettes were filled with either a Cs$^+$-based internal recording solution containing (in mM): 135 CsMeSO$_3$ 10 HEPES, 1 EGTA, 4 Mg-ATP, 0.3 Na-GTP, 8 Na$_2$-Phosphocreatine, 3.3 QX-314 (Cl- salt), pH adjusted to 7.3 with CsOH and diluted to 290–295 mOsm/kg for voltage clamp recordings or a K$^+$-based internal recording solution containing (in mM): 120 KMeSO$_3$, 10 HEPES, 0.2 EGTA, 8 NaCL, 10 KCL, 4 Mg-ATP, 0.3 Na-GTP, pH adjusted to 7.3 with CsOH and diluted to 290–295 mOsm/kg for current clamp recordings.

To stimulate ChR2-expressing neurons, we focused a fiber-coupled 200 mW 473 nm laser (Optoengine) onto the back aperture of 40x Olympus objected in the imaging path. Laser intensity was adjusted using a neutral density filter such that ~ 9 mW/mm$^2$ of total light reached the slice. ChR2$^+$ cells were regularly patched to confirm that laser intensity was well above the threshold needed to elicit action potentials at low latency (data not shown). Cells were classified as having a synaptic response based on the average of at least 10 individual sweeps of optogenetic stimulation. If a

consistent, time-locked response above the baseline noise could be observed, additional sweeps were taken to get a more accurate representation of the response size and kinetics. Putative GABA-mediated currents were isolated by voltage clamping at 0 mV, the reversal potential for excitatory currents, or identified in current clamp as hyperpolarizing potentials that are blocked by 10 µM Gabazine (SR-95531, Tocris). Putative ACh-mediated currents were isolated by voltage clamping at −70 mV, the reversal potential for inhibitory currents, or in current clamp as depolarizing potentials, and confirmed with 10 µM Methyllycaconitine citrate (MLA, Tocris), which is selective for alpha7-containing nicotinic receptors, 10 µM Dihydro-beta-erythroidine hydrobromide (DHBE, Tocris), which is selective for alpha4-containing nicotinic receptors, and 10 µM Mecamylamine hydrochloride (MEC, Tocris), which is a non-selective nicotinic receptor antagonist. To confirm monosynaptic release of GABA, we consecutively added 1 µM TTX (Abcam) followed by 100 µM 4-aminopyridine (Tocris). To rule out contributions from other low latency excitatory receptors, we also added the glutamate receptor antagonists NBQX and CPP (both 10 µM, Tocris). For current-clamp recordings with putative muscarinic receptor-mediated currents, we washed on 10 µM Scopolamine hydrobromide (Tocris), and in a subset of experiments we included 10 µM GCP.-35348 (Tocris) to block GABA$_B$ receptors and 10 µM physostigmine (Tocris) to inhibit ACh esterase.

Voltage clamp and current clamp recordings were amplified and filtered at 3 kHz using a Multiclamp 200B (Axon Instruments) and digitized at 10 kHz with a National Instruments acquisition boards. Data was saved with a custom version of ScanImage written in Matlab (Mathworks; https://github.com/bernardosabatinilab/SabalabSoftware_Nov2009). Additional off-line analysis was performed using Igor Pro (Wavemetrics). Response amplitudes were determined by averaging 5–10 traces, taking a 990 ms baseline prior to stimulation, and subtracting that from the peak amplitude within 5–20 ms after stimulation.

## Fluorescent In situ hybridization
Whole brains dissected from deeply anesthetized wild-type C57/BL6 mice were fresh frozen in Tissue-tek OCT media on dry ice and stored at −80°C before being sliced into 20 µm slices on a CM 1950 Cryostat (Leica), mounted on SuperFrost Plus 25 × 75 mm slides (VWR), and stored at −80°C prior to labeling. Fluorescent in situ hybridization labeling was performed according to the RNAscope Fluorescent Multiplex Assay protocol (ACDBio).

## Immunohistochemistry
Tissue was obtained from deeply anesthetized mice that were perfused transcardially with room temperature phosphate-buffered saline (PBS) followed by 4% paraformaldehyde (PFA) in PBS. The brain was then dissected out of the skull, post-fixed overnight at 4°C in 4% PFA, rinsed and stored in PBS. Brains were sliced into either 50 µm (for most figures) or 25 µm slices (for *Figure 6*) on a Leica VT1000s vibratome and stored in 24-well plates.

For staining, slices were first incubated in blocking buffer (10% Normal Goat Serum, 0.25% Triton-X in PBS, except 10% Normal Horse Serum for ChAT immunostaining) for 1 hr at room temperature on a rotary shaker, then placed in primary antibody solution (1:500 for each primary antibody diluted into carrier solution (10% Normal Goat Serum, 0.2% Triton-X in PBS) and left to shake overnight at 4°C. Slices were then washed 5–6 x in PBS, and placed into secondary antibody solution (1:500 in carrier solution) for 2 hr at room temperature. Slices were again washed, placed on glass slides, and mounted in Prolong Gold antifade mounting media with DAPI (Invitrogen).

## Imaging and analysis
Immunostained and FISH samples were imaged on a VS120 slide scanner at 10x. Regions of interest were then imaged on either a FV1200 confocal microscope (Olympus) or a TCS SP8 confocal microscope (Leica) for colocalization analysis.

Immunostained samples were manually scored to count co-labeled cells using the Cell Counter plugin in Fiji (https://fiji.sc/). FISH samples were analyzed with an automated analysis pipeline custom written using Fiji and Matlab. A cellular mask was created by combining the 3 FISH channels and using the Renyi entropy thresholding algorithm to binarize the image. Each individual cell was identified, and the percent coverage of each FISH channel was calculated for each cell. A threshold to classify each cell as positive or negative for each FISH channel was then determined by selecting a

threshold for percent coverage above ten manually-drawn background areas. From this analysis, we determined the proportion of cells that are positive for each of the different FISH probes.

Immunostained samples from *Figure 6* were imaged using the TCS SP8 confocal microscope (Leica) such that each acquisition utilized the full dynamic imaging range. For analysis, putative individual pre-synaptic terminals were identified by thresholding the raw image stacks of the Synaptophysin-mCherry signal, then filtering putative terminals for size and enforcing that they must be present across multiple images. Mean fluorescence intensity for VGAT and VAChT antibody staining was calculated for each putative terminal. Individual terminals were classified as VGAT or VAChT positive by automatically determining a threshold for VGAT/VAChT positive pixels using the Otsu method (which determines the intensity threshold that minimizes intraclass variance and maximizes interclass variance), and requiring that the terminal is positive or negative if the mean intensity is greater than or equal to the Otsu threshold.

## Array tomography

Brains from mice injected with AAV(8)-CMV-DIO-Synaptophysin-YFP were perfused, dissected, and fixed as for immunohistochemistry. 300 μm thick slices were then cut with a Lieca VT1000s vibratome. Areas of high Synaptophysin-YFP expression were noted using an epifluorescence microscope, and approximately 1 × 1 mm squares of tissue were cut out under a dissecting scope with Microfeather disposable ophthalmic scalpels. These small tissue squares were then dehydrated with serial alcohol dilutions and infiltrated with LR White acrylic resin (Sigma Aldrich L9774), and placed in a gel-cap filled with LR White to polymerize overnight at 50°C. Blocks of tissue were sliced on an ultramicrotome (Leica EM UC7) into ribbons of 70 nm sections.

Antibody staining of these sections was performed as previously described (*Saunders et al., 2015b*). Briefly, antibodies were stained across multiple staining sessions, with up to three antibodies stained per session, and a fourth channel left for DAPI. Typically, Session 1 stained against YFP (chicken α-GFP, GTX13970, GeneTex), Gephyrin (mouse α-Gephyrin, 612632, Biosciences Pharmingen), and Synapsin-1 (rabbit α-Synapsin-1, 5297S, Cell Signaling Tech), Session 2 for PSD-95 (rabbit α-PSD95, 3450 Cell Signaling Tech.), Session 3 for VGAT (rabbit α-VGAT, 131 011 Synaptic Systems), Session 4 for VAChT (mouse α-VAChT, 139 103 Synaptic Systems) and VGLUT1 (guinea pig α-VGAT, AB5905 Millipore), and Session 5 for ChAT (goat α-ChAT, AB144P Millipore). One test sample was performed where the staining order was reversed, and while staining quality did appear degraded for later samples, it was not significant enough to alter analysis. Each round of staining was imaged on a Zeiss Axio Imager upright fluorescence microscope before the tissue ribbons were stripped of antibody and re-stained for a new session of imaging. Four images were acquired with a 63x oil objective (Zeiss) and stitched into a single final image (Mosaix, Axiovision). Image stacks were processed by first aligning in Fiji with the MultiStackReg plug-in, first on the DAPI nuclear stain, with fine alignments performed using the Synapsin 1 stack. Fluorescence intensity was also normalized across all channels, such that the top and bottom 0.1% of fluorescence intensities were set to 0 and maximum intensity, respectively.

For analysis, Synaptophysin-YFP masks were created by first masking out the edges of the images that did not contain any tissue sample and the DAPI signal to exclude cell nuclei, then by empirically determining an appropriate threshold of YFP fluorescence. Putative pre-synaptic terminals were required to exist on multiple z-places of the image stack, thus creating 3D binary masks corresponding to putative pre-synaptic terminals. Global cross-correlations were made by z-scoring the fluorescence signals of each antibody stack making pairwise comparisons among all stacks, shifting the images +/- 10 pixels vertically and horizontally and calculating the 2D co-variance at every shift. We interpreted correlations with DAPI as a proxy measure for the specificity of each pre-synaptic antibody, as these antibodies should completely avoid cell nuclei. In general the antibodies for synaptic markers were excluded from cell nuclei, although VGLUT1, VGAT, ChAT, and VAChT did show small positive correlations with DAPI (*Figure 4D*). Across pairs of pre-synaptic markers, the strongest cross-correlations occurred between Synapsin-1, PSD-95, and VGLUT1, reflecting both the high density of excitatory synapses and relatively low background signal with these antibodies. To specifically analyze co-variance of antibodies within the pre-synaptic terminals, we repeated the calculation of 2D covariance described above, but limited to the area of the images covered by VCIN-expressed Synaptophysin-YFP (~0.1% of the total). Thus the co-expression of synaptic markers within these terminals contributes minimally to the global cross-correlations reported above. To avoid amplifying

any small background signals that would result if an antibody signal was low in the YFP$^+$ pre-synaptic terminals, we z-scored the fluorescence intensities across the entire image stack (as for the global cross correlation analysis above) but calculated the co-variance across signal pairs only within the YFP$^+$ terminals.

Colocalization analysis was carried out using the same YFP mask as described above. Synaptic antibody signals were assigned to individual pixels by fitting each antibody punctum with a Gaussian distribution, and assigning the pixel corresponding to the peak of that Gaussian as the location of that antibody. Colocalization was then calculated by dividing the number of antibody pixels that overlapped with the YFP mask by the total number of pixels in the YFP mask. Similar colocalization values were also calculated within expanding single-pixel concentric volumes around each terminal, to compare the antibody colocalization within terminals with the immediately surrounding tissue. Finally, the location of each antibody puncta was randomized 1000 times, avoiding the DAPI masks, and the colocalization within and around the YFP terminals recalculated for each round of randomization. To compare across samples, this colocalization measure was converted to a z-score by subtracting the mean of the randomized data from the actual colocalization, divided by the standard deviation of the randomized data.

### Blood vessel imaging

For surgical implantation of cranial windows, mice were anesthetized with 2–3% isoflurane, given 10 mg/kg ketoprofen as prophylactic analgesic, and 0.3 mg/kg Dexamethasone to limit tissue inflammation. Mice were placed on a heating pad in a stereotaxic frame (David Kopf Instruments) with continuous delivery and monitoring of appropriate isoflurane anesthesia. The skin above the skull was carefully cleared of hair with scissors and depilatory cream (Nair) and sterilized with alternating scrubs with alcohol pads and betadine pads. A midline incision was made in the skin and the skull exposed. A circular,~3 mm diameter section of skull was carefully drilled from over the right barrel cortex, with frequent application of sterile saline. A cranial window, prepared by adhering a 3 mm glass coverslip to a 4 mm coverslip with optical glue, was placed over the brain, and secured in place with Kwik-cast silicone elastomer sealant (World Precision Instruments), followed by C and B-Metabond (Parkell) with a custom-made titanium head post. Following surgery, mice were monitored in their home cage for 4 days following surgery, and received daily analgesia for 2 days following surgery.

Alexa Fluor 633 hydrazide (5 mg•kg$^{-1}$) was retro-orbitally injected into mice to visualize arterioles in vivo. Arterioles were imaged at 800 nm with a field of view size of 200 μm x 200 μm (512 × 512 pixels, pixel size of 0.16 μm$^2$/pixel) at 30 Hz. Optical stimulation was performed using pulsed illumination (5 pulses, 20 Hz, 5 ms ON/45 ms OFF, 30 mW/mm2) using a 473 nm solid-state laser. Whisker stimulation (4 Hz, 5 s) was performed using a foam brush controlled by a servo motor under the control of Wavesurfer. Three technical trials where acquired and averaged for each field of view. 10–13 fields of view were acquired per imaging session. Three imaging sessions were collected on three separate days per mouse and arteriolar dilation responses were averaged across all three sessions for each mouse.

### Acknowledgements

The authors thank V Prado and M Prado for generous donation of the *Slc18a3$^{fl/fl}$* (VAChT flox) mouse line, B Rudy for donation of the 5HT3aR-GFP mouse line, and N Kingery and T Xie for production and analysis of array tomography data. Thanks to members of the Sabatini lab for thoughtful critique of the manuscript. We thank G Fishell for helpful discussions and advice on the project and manuscript. This work was supported by a fellowship from Jane Coffin Childs Fund (AJG), and grants from the NIH (K99 NS102429 to AJG, R37 NS046579 to BLS, and P30NS072030 to the Neurobiology Imaging Facility).

# Additional information

## Funding

| Funder | Grant reference number | Author |
|---|---|---|
| National Institute of Neurological Disorders and Stroke | R37 NS046579 | Bernardo L Sabatini |
| National Institute of Neurological Disorders and Stroke | K99 NS102429 | Adam J Granger |
| National Institute of Neurological Disorders and Stroke | P30Ns072030 | Mahmoud El-Rifai |
| Jane Coffin Childs Memorial Fund for Medical Research | | Adam J Granger |

The funders had no role in study design, data collection and interpretation, or the decision to submit the work for publication.

## Author contributions

Adam J Granger, Conceptualization, Data curation, Formal analysis, Funding acquisition, Investigation, Visualization, Methodology, Writing - original draft, Writing - review and editing; Wengang Wang, Keiramarie Robertson, Mahmoud El-Rifai, Andrea F Zanello, Karina Bistrong, Brian W Chow, Vicente Nuñez, Data curation; Arpiar Saunders, Conceptualization, Data curation; Miguel Turrero García, Corey C Harwell, Chenghua Gu, Data curation, Writing - review and editing; Bernardo L Sabatini, Conceptualization, Resources, Supervision, Funding acquisition, Methodology, Writing - original draft, Writing - review and editing

## Author ORCIDs

Adam J Granger (ID) https://orcid.org/0000-0002-3953-6764
Miguel Turrero García (ID) http://orcid.org/0000-0002-7294-169X
Chenghua Gu (ID) http://orcid.org/0000-0002-4212-7232
Bernardo L Sabatini (ID) https://orcid.org/0000-0003-0095-9177

## Ethics

Animal experimentation: This study was performed in accordance with the recommendations in the Guide for the Care and Use of Laboratory Animals of the NIH, and according to strict adherence to the protocols approved by the Institutional Animal Care and Use Committee (IACUC) of Harvard Medical School (protocol #IS00000571). Routine examination, veterinary care, disease surveillance, and animal use compliance were all carried out by certified veterinary staff of the Harvard Center for Comparative Medicine (HCCM) in addition to full daily animal husbandry provided by trained animal technicians.

## Decision letter and Author response

Decision letter https://doi.org/10.7554/eLife.57749.sa1
Author response https://doi.org/10.7554/eLife.57749.sa2

# Additional files

## Supplementary files

• Transparent reporting form

## Data availability

All data generated during this study are summarized in the figures and supporting files of this manuscript. Source data files from which the figures were generated are available at: https://dataverse.harvard.edu/dataset.xhtml?persistentId=https://doi.org/10.7910/DVN/AIUTNJ.

The following dataset was generated:

| Author(s) | Year | Dataset title | Dataset URL | Database and Identifier |
|---|---|---|---|---|
| Granger A | 2020 | Replication Data for: Cortical ChAT + neurons co-transmit acetylcholine and GABA in a target- and brain-region specific manner | https://dataverse.harvard.edu/dataset.xhtml?persistentId=doi:10.7910/DVN/AIUTNJ | Harvard Dataverse, 10.7910/DVN/AIUTNJ |

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
