## [Decision Letter]

**Acceptance summary:**

This paper combines anatomical and functional approaches to show that interneurons in the neocortex target two neurotransmitters to different sets of cellular targets. In addition, they show that cortical areas contain different subtypes of these interneurons. These results suggest additional mechanisms by which local interneurons regulate the patterns of neuronal activity in the neocortex in a cell-type and region-specific manner.

**Decision letter after peer review:**

Thank you for submitting your article "Cortical ChAT^+^ neurons co-transmit acetylcholine and GABA in a target-and brain-region specific manner" for consideration by *eLife*. Your article has been reviewed by three peer reviewers, and the evaluation has been overseen by a Reviewing Editor and Kenton Swartz as the Senior Editor. The following individuals involved in the review of your submission have agreed to reveal their identity: Chris J McBain (Reviewer #2); Steven Shabel (Reviewer #3).

The reviewers have discussed the reviews with one another, and the Reviewing Editor has drafted this decision to help you prepare a revised submission.

We would like to draw your attention to changes in our revision policy that we have made in response to COVID-19 (https://elifesciences.org/articles/57162). Specifically, when editors judge that a submitted work as a whole belongs in *eLife* but that some conclusions require additional analyses of the data, as they do with your article, we are asking that the manuscript be revised to preferably include these additional analyses or, when that is not possible under the current conditions, to either limit claims to those supported by data in hand or to explicitly state that the relevant conclusions require additional supporting data.

In this article, Granger et al. use optogenetic stimulation to examine the functional connectivity of VIP^+^/ChAT^+^ cortical interneurons in brain slices. Surprisingly, despite extensive expression of cholinergic markers in presynaptic cortical terminals, they observe few fast, cholinergic responses across cortical layers and cell types in primary motor cortex (M1). In contrast, they find widespread GABAergic transmission (predominantly in SOM^+^ interneurons), consistent with the expression of GABAergic markers in VIP^+^/ChAT^+^ neurons. Although few cells show clear ACh only transmission, the authors confirm using array tomography that presynaptic axon terminals can segregate ACh and GABA content, showing that release of ACh or GABA may be regulated between distinct downstream targets. To complement their studies in M1, the authors tackle the previous observation of ACh release from ChAT^+^ neurons in prefrontal cortex (PFC), data which would argue against their primary observation in motor cortex. They show evidence for a rare, VIP-/ChAT^+^ interneuron found in mPFC but not M1 and provide data implicating these in the more commonly detected ACh responses seen in PFC, highlighting circuit distinctions between these two cortical regions.

Revisions expected for this paper:

Overall, all three reviewers agreed that the experiments were well designed and well executed. One major concern related to the lack of explicit comparisons between M1 and PFC in some experiments (like the array tomography experiments) and clarifying the laminar and areal locations of other experiments as these areal and laminar differences are presented as major findings of the paper. Additional analyses explicitly comparing array tomography images between M1 and PFC and from layer 1 and other layers (laminar distribution) would significantly strengthen the manuscript. Furthermore, reporting, for example, the layer breakdown of the recordings from inhibitory neurons in Figure 3G and the layer and areal locations of the data analyzed in Figures 4 and 5 would improve the manuscript. Reporting the data for M1 and PFC separately in Figure 6J and indicating how many neurons were recorded in both regions would also contribute to these comparisons. A second major concern was that some experiments lacked quantification or explicit descriptions of the quantification. For example, additional quantification such as reporting the proportion of ChAT^+^ neurons that are VIP- (Figure 7) and the proportion of ChAT^+^ cells that are Nkx.2.1-flp/ChAT-cre/RC::FLTG ones would improve the manuscript. As another example, in Figure 3G, the statistical test and p value to demonstrate that SOM^+^ neurons show more frequent responses and the number of mice contributing to the data should be specified. For all experiments, the number of sections or cells analyzed, the number of mice contributing to the data, the statistical test and p values and so on should be clearly stated.

Revisions for this paper or a follow-up:

Working on the assumption that new experiments are not possible under current conditions, any existing experimental results testing the effects of uptake blockers or focused on mAChRs would also add to the manuscript. Did the authors attempt to use uptake blockers and or explore muscarinic versus nicotinic responses in any detail to determine if point-to-point transmission is not the main means of communication of ACh? The punctate and presynaptic terminal expression would argue against this, but the small amplitude of currents detected by optogenetics suggests that this may not be a primary mode of transmission. Similarly, in paragraph three of subsection “Cortical VIP^+^/ChAT^+^ neurons robustly release GABA onto inhibitory interneurons and sparsely

release ACh” the authors suggest that a subtype of layer 1 neurons receives ACh responses. As there are distinct types of layer 1 interneurons (e.g. Schuman et al., 2019), any electrophysiological and/or morphological phenotyping of the postsynaptic cell type would provide more compelling evidence for the conclusion, that a highly specific subnetwork in L1 is targeted by VIP/ChAT interneurons.

---

## [Author Response]

Revisions expected for this paper:Overall, all three reviewers agreed that the experiments were well designed and well executed. One major concern related to the lack of explicit comparisons between M1 and PFC in some experiments (like the array tomography experiments) and clarifying the laminar and areal locations of other experiments as these areal and laminar differences are presented as major findings of the paper.

We have attempted to address these concerns by, whenever possible, including such information. We now note the areal locations of the analyses we are performing, and added additional accounting of laminar positions of all cells we recorded from, and where possible, of the pre-synaptic terminals we characterize.

Nevertheless, we do not believe laminar differences between pre-synaptic terminal neurotransmitter expression or synaptic connectivity explain the differences in ACh vs. GABA release. Layer 1 interneurons showed among the highest rates of ACh connectivity, but we think this is a feature of layer 1 interneurons as a cell type, and not as a specific laminar enrichment of ACh release. Layer 1 is unique in that it lacks pyramidal neurons, and for the most part the SOM^+^ and PV^+^ interneurons that span the other cortical layers.

Additional analyses explicitly comparing array tomography images between M1 and PFC and from layer 1 and other layers (laminar distribution) would significantly strengthen the manuscript.

Unfortunately, given the shutdown of all lab operations and core facilities as a result of the COVID-19 pandemic, we are currently unable to perform any additional array tomography experiments. Even as lab operations ramp back up, it will be an unknown amount of time until we are able to make use of core facility services that we would require for additional array tomography experiments.

Beyond this logistical constraint, there are other practical concerns that make these experiments infeasible. Each image stack of array tomography data only covers ~200 x 250 microns of cortical tissue. Therefore, in order to compare array tomography between layers, spanning the entire cortical column across multiple replicates, would require dramatically scaling up the throughput of our array tomography data acquisition beyond the capabilities of our facility. To our knowledge, such large-scale, high-throughput array tomography experiments are rare outside of the Allen Institute.

Additionally, we are dubious of the specific hypothesis answered by comparing array tomography between M1 and mPFC. Our array tomography experiments are meant to specifically address the VIP^+^/ChAT^+^ interneuron population. We show that the connectivity difference between M1 and mPFC are due to the presence of non-VIP, ChAT^+^ neurons in the mPFC. Therefore any comparison of VIP^+^/ChAT^+^ pre-synaptic terminals between mPFC and M1 with array tomography would require intersectional genetics to disambiguate the contribution of these two cell types, without any evidence a priori for a difference within VIP^+^/ChAT^+^ interneurons across brain regions.

However, we do have data pertaining to the laminar distribution of VAChT and VGAT-expressing terminals from VIP^+^/ChAT^+^ neurons in motor cortex from the immunohistochemistry experiments described in Figure 5. We have included this additional analyses in Figure 5—figure supplement 1. Contrary to expectations from the synaptic connectivity data, we actually find a very slight (but significant) depletion of VAChT^+^ terminals in superficial layers, and enrichment of VGAT^+^ terminals. However, because the position of pre-synaptic terminals is not that informative for the position of the cells that receive those synapses (neurons of deep layers receive inputs onto their dendrites in layer 1), we do not draw any strong conclusions from these laminar distribution analysis. Still, we hope these additional analyses is helpful towards addressing the reviewers’ concerns.

Furthermore, reporting, for example, the layer breakdown of the recordings from inhibitory neurons in Figure 3G and the layer and areal locations of the data analyzed in Figures 4 and 5 would improve the manuscript.

We have added the laminar positions of all recordings from 3G as supplemental data in Figure 3—figure supplement 1, as well as the areal locations of the all data for Figures 4 and 5 as indicated in the text and figure legends, and laminar distribution of all terminals from Figure 5 are indicated in Figure 5—figure supplement 1.

Reporting the data for M1 and PFC separately in Figure 6J and indicating how many neurons were recorded in both regions would also contribute to these comparisons.

We have changed Figure 6 to report the data separately for M1 and mPFC in Figure 6J as suggested, indicating the number of neurons recorded in both regions. Our reporting of cells recorded in gabazine has been moved to Figure 6—figure supplement 2.

A second major concern was that some experiments lacked quantification or explicit descriptions of the quantification. For example, additional quantification such as reporting the proportion of ChAT^+^ neurons that are VIP- (Figure 7) and the proportion of ChAT^+^ cells that are Nkx.2.1-flp/ChAT-cre/RC::FLTG ones would improve the manuscript.

We have aimed to provide quantification wherever possible in the manuscript, but the uniquely sparse nature of the Nkx2.1/ChAT neurons (which are consistent with being the same as the non-VIP CHAT^+^ neurons) makes this difficult – as only a small handful are present in any given mouse brain. Thus, for the whole brain the answer is going to be something like 3-5 VIP-/Chat+ cells compared to thousands of VIP+/Chat+ cells. If we analyze only the fields of view in which we find VIP-/Chat+ cells, this will erroneously give the impression that they are a high fraction of cells.

Therefore, we thought the most clear and correct way to describe the prevalence of these neurons is to merely report that we often did not find more than 3-5 in each mouse brain. We have also edited the text to make more clear that the non-VIP ChATs we observe through FISH and the genetically labeled Nkx2.1/ChAT-expressing neurons are similarly sparse and we believe to be the same population.

As another example, in Figure 3G, the statistical test and p value to demonstrate that SOM^+^ neurons show more frequent responses and the number of mice contributing to the data should be specified. For all experiments, the number of sections or cells analyzed, the number of mice contributing to the data, the statistical test and p values and so on should be clearly stated.

We have now included the number of cells and mice contributing to each data set in the paper. Unfortunately, we do not have the exact number of slices for all physiology experiments.

We have also added Pearson’s Chi-squared statistical p-values to report the likelihood that the differences in connectivity we observe in Figure 3 occurred by chance, and have added those p-values to the figure legend.

Revisions for this paper or a follow-up:Working on the assumption that new experiments are not possible under current conditions, any existing experimental results testing the effects of uptake blockers or focused on mAChRs would also add to the manuscript. Did the authors attempt to use uptake blockers and or explore muscarinic versus nicotinic responses in any detail to determine if point-to-point transmission is not the main means of communication of ACh? The punctate and presynaptic terminal expression would argue against this, but the small amplitude of currents detected by optogenetics suggests that this may not be a primary mode of transmission.

We did perform extensive experiments to detect mAChR-mediated responses by stimulating with trains of blue light while recording in K^+^-internal solutions. We recorded over 200 hundred neurons, some in the presence of GABA-receptor antagonists (gabazine and CGP-35348) and ACh-esterase inhibitors (physostigmine) in order to boost our chances of observing ACh-mediated responses. These experiments were done using the *VIP^ires-Cre^ x Rosa26^lsl-ChR2-EYFP^* mouse line and have been added to the manuscript in Figure 6—figure supplement 2. One confusing part of this analysis is that our optogenetic stimulation often caused a depolarization that could be mistaken for mAChR-mediated responses. However, we found that these voltage changes could not be consistently blocked by ACh-receptor antagonists. They were blocked by TTX, but not rescued by 4AP. We also observed that they could not be blocked by Cd^2+^, indicating that these depolarization are not caused by synaptic release. We therefore concluded that these were ephaptic depolarizations occurring as a result of strongly activating VIP interneuron axons. As these experiments do not add to our major findings, we have limited our explanation of these results to the Figure 5—figure supplement 1 figure legend.

Similarly, in paragraph three of subsection “Cortical VIP^+^/ChAT^+^ neurons robustly release GABA onto inhibitory interneurons and sparsely release ACh” the authors suggest that a subtype of layer 1 neurons receives ACh responses. As there are distinct types of layer 1 interneurons (e.g. Schuman et al., 2019), any electrophysiological and/or morphological phenotyping of the postsynaptic cell type would provide more compelling evidence for the conclusion, that a highly specific subnetwork in L1 is targeted by VIP/ChAT interneurons.

Unfortunately, we do not have any further electrophysiological or morphological information about those layer 1 interneurons. We agree this data would be interesting, but electrophysiological characterization is not possible using the Cs^+^-based internal we used for screening nAChR-mediated responses. In the future we hope to perform morphological reconstruction of connected neurons.